# CROSSLINGUAL REASONING THROUGH TEST-TIME SCALING

## ABSTRACT

Reasoning capabilities of large language models are primarily studied for English, even when pretrained models are multilingual. In this work, we investigate to what extent English reasoning finetuning can generalize across languages. First, we find that sequential test-time scaling for English-centric reasoning language models (RLMs) through longer chain-of-thoughts (CoTs) improves multilingual mathematical reasoning across many languages including low-resource languages, to an extent where they outperform models *twice their size*. Second, we reveal that while English-centric RLM's CoTs are naturally predominantly English, they consistently follow a *quote-and-think* pattern to reason about quoted non-English inputs. Third, we discover an effective strategy to control the language of long CoT reasoning, and we observe that models reason better and more efficiently in high-resource languages. Overall, we demonstrate the potentials, study the mechanisms, and outline the limitations of crosslingual generalization of English reasoning test-time scaling. We conclude that practitioners should let English-centric RLMs reason in high-resource languages, while further work is needed to improve reasoning in low-resource languages.

## 1 INTRODUCTION

Scaling up compute at test-time can maximize model performance and output quality (Snell et al., 2024; Brown et al., 2024; Wu et al., 2024; Levi, 2024), but it has been understudied in multilingual settings. In particular, reasoning language models (RLMs), such as Deepseek's R1 (Guo et al., 2025) and OpenAI's o1 or o3 models (Jaech et al., 2024; OpenAI, 2025), strongly benefit from added inference compute to their long chain-of-thoughts (long CoTs) (Chen et al., 2025), also known as *sequential test-time scaling*. However, this advantage has primarily been explored in English contexts, such as in recent work that combined small-scale reasoning finetuning with scaled up number of thinking tokens at test time (Muennighoff et al., 2025; Ye et al., 2025). State-of-the-art RLMs rely on reasoning training data that contain long CoTs, which is currently most available for English (Ghosh et al., 2025). Thus, these RLMs are English-centric (Muennighoff et al., 2025; Hou et al., 2025; Hao et al., 2024; Gou et al., 2024; Xiang et al., 2025; Ghosh et al., 2025). Given that their base models are often multilingual models such as Qwen (Yang et al., 2024), does reasoning finetuning in English give them multilingual reasoning abilities?

In this work, we investigate *how much test-time compute can improve multilingual reasoning abilities of English-centric RLMs*. In particular, our research questions are as follows:

RQ1. **Crosslingual test-time scaling**: How effective is test-time scaling of English-centric RLMs on multilingual reasoning tasks? (Section 4)

RQ2. **Language-mixing behaviors**: What kind of language-mixing patterns do English-centric RLMs exhibit when they interact with non-English prompts? (Section 5)

RQ3. **Language forcing**: How well do English-centric RLMs perform when being forced to think in non-English languages? (Section 6)

We experiment with s1 models (Muennighoff et al., 2025) as our English-centric RLMs for crosslingual generalization study. They are multilingual Qwen2.5-Instruct models (Yang et al., 2024)

supervised finetuned on 1k training samples of English STEM reasoning tasks and achieve state-of-the-art performance on English math reasoning benchmarks (Muennighoff et al., 2025). Our most significant contributions are as follows:

1. We provide evidence that larger models benefit from crosslingual test-time scaling that extends CoT, which *contrasts* with contemporary work (Son et al., 2025) that draws negative conclusions based on 1.5B models. Crosslingual test-time scaling is not only effective for both high-resource and low-resource languages across different difficulty levels, but it can even allow an RLM to outperform models twice its size on multilingual math reasoning tasks.

2. We report a dominant language-mixing pattern where RLMs quote non-English phrases related to the question prompts in quotation marks in the thinking process. This *quote-and-think* pattern suggests that model's multilingual capability to parse and understand questions enables crosslingual generalization of English reasoning finetuning.

3. We discover an effective strategy to control the reasoning language of RLMs, and we find that forcing RLMs to think in high-resource languages yields substantially better reasoning performance than in low-resource languages. Furthermore, the long CoTs for high-resource languages are more token-efficient at test time.

Our work shows that test-time scaling of English-centric RLMs can serve as a strong multilingual reasoning baseline. We recommend letting the English-centric RLMs reason in high-resource languages such as English and Chinese for optimal performance and inference-compute efficiency. Future work is needed for enabling RLMs to generalize to better reason in low-resource languages.

## 2 BACKGROUND AND RELATED WORK

**Reasoning language models (RLMs).** Recent advancements of reasoning language models (RLMs) such as OpenAI-o1 (Jaech et al., 2024; OpenAI, 2025) and Deepseek-R1 (Guo et al., 2025) builds on LLMs' capability to perform intermediate reasoning steps, which is commonly referred to as chain-of-thought reasoning (Wei et al., 2022). Prior work demonstrates that these intermediate computation steps can significantly improve the correctness for final answer outputs (Wei et al., 2022; Ling et al., 2017; Cobbe et al., 2021; Nye et al., 2021; Li et al., 2024b). Furthermore, extending the lengths of these computation steps, thereby creating long chain-of-thoughts (long CoTs), can allow the model to backtrack on incorrect reasoning steps and self-correct its final answer (Chen et al., 2025; Gandhi et al., 2025; Guo et al., 2025; Hou et al., 2025; Lee et al., 2025). In our work, we focus on RLMs with long CoTs capability, which is an emerging research area. These models are created through *distilling* long English-only reasoning chains from larger RLMs (Huang et al., 2024b; Ye et al., 2025; Labs, 2025; Madhusudhan et al., 2025; Wang et al., 2025a) to finetune multilingual pretrained models like Qwen models (Yang et al., 2024); yet, there is a limited understanding on how pretrained models' multilingual capability enables crosslingual reasoning of long CoTs.

**Sequential test-time scaling and s1.** Sequential test-time scaling is a new scaling paradigm where more computation budget is allocated for model generation lengths g before committing to an answer (Snell et al., 2024). This is usually done by scaling up the number of thinking tokens during CoT (Snell et al., 2024; Goyal et al., 2024; Jaech et al., 2024; Guo et al., 2025), which contrasts parallel scaling that samples multiple generations and pick the best one (Khairi et al., 2025; Snell et al., 2024). The s1 work (Muennighoff et al., 2025) demonstrates the effectiveness of a simple sequential test-time scaling recipe: reasoning finetuning on small amount of training data with long CoTs (specifically 1k samples distilled from larger RLMs such as Deepseek-R1) and scaling up inference budget at test time. Through test-time scaling of a 32B-parameter model, the authors achieve the state-of-the-art mathematical reasoning performance, and their models even rival industry-grade RLMs such as o1-mini (Jaech et al., 2024). Nonetheless, similar to aforementioned RLMs literature, exploration of test-time scaling paradigm mostly evaluate on English math benchmarks (Snell et al., 2024; Ghosh et al., 2025; Levi, 2024; Xiang et al., 2025; Wu et al., 2024; Muennighoff et al., 2025). Here, our work focuses on understanding how effective test-time scaling of English-centric RLMs, specifically s1 models, in multilingual settings.

**Multilingual reasoning.** Multilingual reasoning encompasses the ability of language models to perform complex reasoning tasks across different languages. Early work has demonstrated that chain-of-thought prompting in English can significantly improve performance on multilingual mathematical

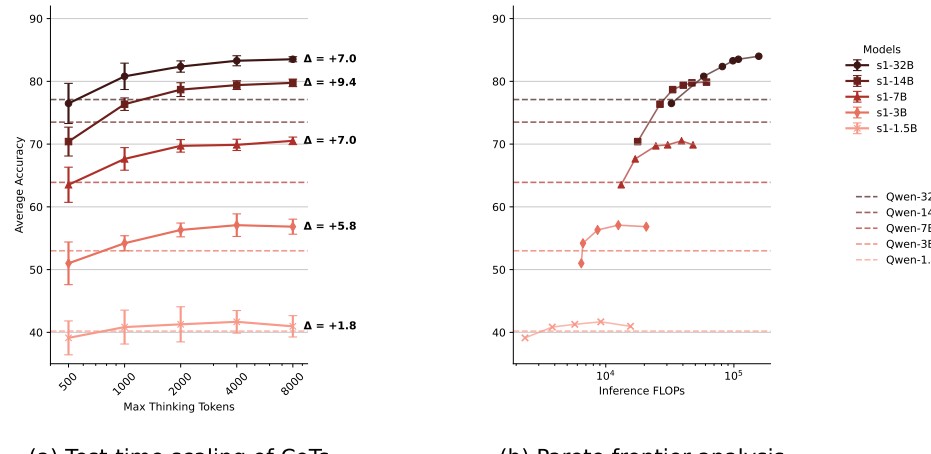

(a) Test-time scaling of CoTs          (b) Pareto frontier analysis

Figure 1: Crosslingual test-time scaling of s1 and Qwen models on the MGSM benchmark (*excluding English*) across different model sizes. In subfigure (a) we enforce a hard limit of maximum thinking token, and in (b) we measure their inference FLOP compute for a Pareto frontier analysis. $\Delta$ measures the absolute difference between average accuracy scores at 0.5k and 8k maximum thinking tokens. Dash lines indicate the best CoT prompting baseline performance of Qwen.

reasoning tasks (Shi et al., 2023), which suggests that LLMs might rely on dominant languages like English as a pivot language for complex reasoning. Follow-up work explores several strategies such as translating the multilingual queries to English (Qin et al., 2023; Zhu et al., 2024; Ko et al., 2025), aligning of latent representation spaces (Yoon et al., 2024; Huang et al., 2024c) and reasoning outputs (She et al., 2024; Yang et al., 2025; Ranaldi & Pucci, 2025; Gao et al., 2025) across languages, or expanding language coverage of reasoning training data or in-context examples (Chen et al., 2023; Li et al., 2024a; Tu et al., 2025). Nonetheless, some work reports opposite findings on whether English is the best pivotal language (Turc et al., 2021).

Our work focuses on understanding how controlling the length of long CoTs and their reasoning language at test time affects multilingual reasoning. One similar work (Son et al., 2025) experimented with controlling generation lengths of finetuned Deepseek-R1-1.5B (Guo et al., 2025) but reported *negative* results: increasing thinking tokens leads to minimal performance gains for mathematical reasoning in non-English languages. We believe that their negative findings are due to constrained model parameters, as we show that larger models *can benefit* from crosslingual test-time scaling.

## 3 EXPERIMENTAL SETUP

**Models.** We use s1 models (Muennighoff et al., 2025) as our English-centric RLMs. We work with the s1.1 variants, which are multilingual Qwen2.5-Instruct models finetuned on 1k English-only reasoning data generated by Deepseek-R1. Both the model weights and training data of s1 are fully open-sourced. We experiment with s1 models at different scales, namely 1.5B, 3B, 7B, 14B, and 32B parameters.

**Budget forcing.** Budget forcing refers to techniques for controlling inference budget for long CoTs (Muennighoff et al., 2025), which can be done in two ways: (1) **truncation**, which *cuts off* long CoTs after they reach maximum thinking tokens, or (2) **extrapolation**, which *adds* tokens such as "Wait" at the end of CoTs to force the model continue reasoning. In our extrapolation setup, we follow (Muennighoff et al., 2025) and experiment with adding "Wait" at the end of s1's thinking to lengthen its CoTs.

**Evaluation data.** We use the following three multilingual math reasoning benchmarks:

1. **MGSM** (Shi et al., 2023): A benchmark that contains 250 grade-school math problems manually translated from the GSM8K dataset into ten languages, namely Bengali (`bn`), German (`de`),

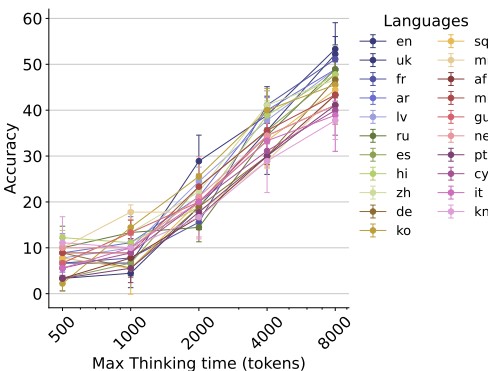
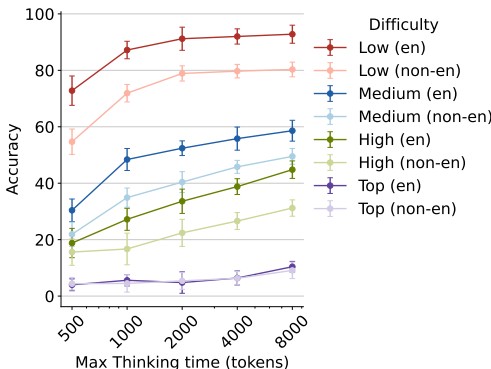

Figure 2: Performance of s1-32B on MT-AIME2024 benchmark across 21 languages. Each data point is an average result of 16 different seeds.

Figure 3: Performance of s1-32B on PolyMath benchmark across four difficult levels (i.e., low, medium, high, and top). We average the results of non-English languages.

Spanish (`es`), French (`fr`), Japanese (`ja`), Russian (`ru`), Swahili (`sw`), Telugu (`te`), Thai (`th`), and Mandarin Chinese (`zh`).

2. **MT-AIME2024** (Son et al., 2025): A benchmark that contains 30 machine-translated math problems from American Invitational Mathematics Examination (AIME) 2024 (MAA, 2024). We evaluated on 21 languages covering both high-resource and low-resource languages including Kannada (`kn`), Guarani (`gn`) and Marathi (`mr`). This dataset is more challenging than MGSM.

3. **PolyMath** (Wang et al., 2025b): A benchmark that contains 500 math problems in total at four different difficulty levels from K-12 math level (i.e., low level) to Olympiads problems (i.e., top level). The questions are directly sourced from school exams and math competitions and are translated into 18 languages by language experts.

We use the `lm-evaluation-harness` library (Gao et al., 2024) as the main evaluation framework. We sample our outputs using 5 different random seeds, temperature of 0.6 and top-p of 0.95, and we report task accuracy, which is equivalent to pass@1.

## 4 CROSSLINGUAL TEST-TIME SCALING

In `RQ1`, we explore test-time scaling in a *zero-shot crosslingual setting*, where English-centric reasoning models are applied to math problems in different languages.

### 4.1 EFFECTIVENESS OF CROSSLINGUAL TEST-TIME SCALING

**Crosslingual generalization of reasoning training and test-time scaling.** We report two main obsevations from Figure 1 (a). First, we observe that s1 outperforms Qwen's few-shot prompting baseline across languages in MGSM (excluding English) when given high inference thinking budget. Second, crosslingual test-time scaling is effective for models with 3B parameters and above. We want to highlight that *sufficient model capacity* is necessary for effective crosslingual test-time scaling, as test-time scaling only yields minimal benefits (only $+\Delta 1.8\%$) at 1.5B size.

Figure 1 (b) illustrates the performance-efficiency trade-off across different sizes of the s1 model family. We follow prior test-time scaling work (Snell et al., 2024; Sardana et al., 2023) and compute the inference cost using the approximation FLOPs $= 2ND_{\text{inference}}$ where $N$ represents model parameters and $D_{\text{inference}}$ the total number of tokens generated at inference time, and we average across different languages. The figure further demonstrates the model capacity constraint on test-time scaling, as we have not observed any performance surge with more compute for smaller models in our experiments that matches the performance by 32B and 14B models. While we observe accuracy-to-computation tradeoffs (i.e., better performance comes with using larger models and higher test-time compute), the 14B model offers a compelling compromise by achieving above 80%

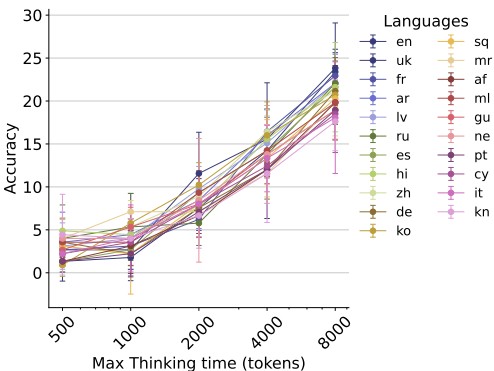
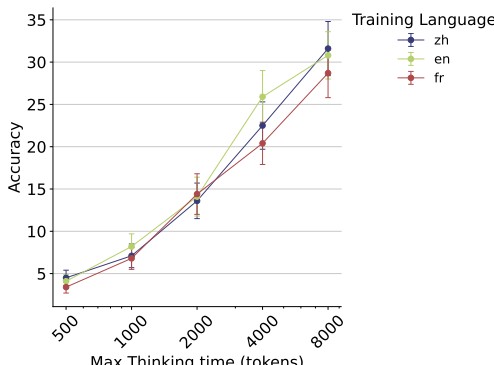

Figure 4: Performance of Llama3.1-8B-Instruct, trained with s1.1 dataset, on MT-AIME2024 benchmark.

Figure 5: Performance of s1-14B, trained on different high-resource reasoning language, on the low-resource languages of the MT-AIME2024 benchmark.

accuracy with substantially lower inference FLOPs than the 32B model on the easier benchmark MGSM, representing a "sweet spot" on the Pareto frontier for practical applications.

**Crosslingual generalization at different difficulty levels.** Figure 2 demonstrates similar findings with Figure 1 (a) on the harder benchmark MT-AIME2024. Across all 21 languages tested, including both high-resource languages like German (de) and French (fr) and low-resource languages like Guarani (gn) and Kannada (kn) and Guarani, s1-32B shows consistent improvements from test-time scaling, with task accuracy increases drastically from mostly below 10% to above 40% as thinking tokens increase from 500 to 8000. In Appendix B.3, we analyze the possible linguistic factor that contributes to crosslingual reasoning transfer.

To further understand how problem complexity affects crosslingual generalization, we analyze performance on the PolyMath benchmark. We want to note that the most of the math reasoning data for s1 (Muennighoff et al., 2025) is at "high" difficulty level according to the difficulty classification scheme by Wang et al. (2025b). Figure 3 shows that crosslingual test-time scaling consistently improves performance across all four difficulty levels for both English and non-English languages, though the effectiveness varies considerably by complexity.

Most notably, at the top difficulty level representing Olympic-level problems, where s1 reasoning training does not readily transfer to solving more challenging questions, both language groups struggle significantly with English performance dropping to around 10% and non-English languages performing even lower at roughly 5%. Nonetheless, for difficulty levels within s1's competence range (low, medium, and high), crosslingual test-time scaling demonstrates robust effectiveness with substantial performance gains from increased thinking tokens.

## 4.2 PERFORMANCE COMPARISON ON MGSM BENCHMARK

We break down performance of s1-14B against other models on the MGSM benchmark. Since MGSM comes with training data, we can perform few-shot prompting on the Qwen base model following (Shi et al., 2023) by prompting either in English (EN-CoT) or in the same language as input prompt (native-CoT). In addition, we benchmark against prior state-of-the-art models that are trained on *multilingual* reasoning data to solve MGSM (Yue et al., 2023; She et al., 2024; Fan et al., 2025; Yoon et al., 2024; Team et al., 2025). For performance comparison on PolyMath benchmark, we refer readers to Appendix B.2.

**Comparison against Qwen baselines.** Table 1 shows that with crosslingual test-time scaling, s1 gains substantial accuracy increase as compared to different baselines with Qwen2.5 models. Furthermore, crosslingual test-time scaling benefits *both* high-resource and low-resource languages. For instance, fr receives a significant $+\Delta 23.1\%$ relative accuracy increase, whereas sw—the worst-performing language for the base model Qwen—receives $+\Delta 41.6\%$ relative accuracy improvement. Lastly, we observe similar performance for both truncation and extrapolation budget forcing strategies. This is

Table 1: MGSM performance comparison against 14B-sized s1 model with maximum 8k thinking tokens. We report the language-breakdown accuracy from cited papers if available; otherwise, we reproduce using their open-sourced models without any inference budget constraint. We report the average length of the generations (avg. len) and the relative accuracy difference (green text) between s1-14B under extrapolation budget forcing and its baseline Qwen2.5-14B-Instruct. We **bold** both s1 performance and baseline models that outperform s1.

| Models | avg len | bn | de | en | es | fr | ja | ru | sw | te | th | zh | AVG |
|---|---|---|---|---|---|---|---|---|---|---|---|---|---|
| Qwen2.5-14B-Instruct (Yang et al., 2024) | 413.1 | 74.0 | 77.6 | 82.0 | 77.6 | 67.6 | 70.4 | 76.4 | 40.4 | 50.8 | 78.8 | 84.0 | 70.9 |
| + 8-Shot EN-CoT Shi et al. (2023) | 316.5 | 77.2 | 75.2 | 87.6 | 86.0 | 68.4 | 76.8 | 76.4 | 45.6 | 52.0 | 79.2 | 84.4 | 73.5 |
| + 8-Shot Native-CoT Shi et al. (2023) | 365.2 | 79.2 | 77.2 | 88.0 | 87.2 | 68.4 | 76.0 | 75.6 | 46.8 | 53.2 | 80.4 | 83.6 | 74.1 |
| s1-14B (truncation) | 1912.9 | 82.0 | 84.8 | **92.8** | **88.4** | 85.2 | 83.6 | 86.8 | 55.6 | **59.6** | 85.2 | 86.4 | 80.9 |
| **s1-14B (extrapolation)** | 2352.3 | **82.8** | **86.8** | 92.4 | 86.4 | 83.2 | 83.2 | **88.8** | **57.2** | 58.0 | 84.8 | **87.6** | **81.0** |
| Relative accuracy difference (%) | | +11.9% | +11.9% | +12.7% | +11.3% | +23.1% | +18.2% | +16.2% | +41.6% | +14.2% | +7.6% | +4.3% | +14.2% |
| MetaMath-13B (Yu et al., 2024) | 529.8 | 6.8 | 64.4 | 70.4 | 63.6 | 65.2 | 47.6 | 60.0 | 11.6 | 0.8 | 4.8 | 50.8 | 40.5 |
| MetaMathOctopus-13B She et al. (2024) | 545.8 | 41.6 | 60.1 | 66.8 | 61.1 | 60.8 | 57.3 | 59.1 | 50.9 | 3.6 | 52.1 | 53.1 | 51.5 |
| MAPO-DPO-13B She et al. (2024) | 552.4 | 54.7 | 69.5 | 70.5 | 70.6 | 71.3 | 69.0 | 68.2 | 62.9 | 4.0 | 64.7 | 68.2 | 61.2 |
| SLAM-13B Fan et al. (2025) | 101.5 | 45.6 | 62.8 | 71.2 | 67.6 | 65.2 | 54.0 | 64.4 | 46.4 | 2.4 | 47.6 | 58.8 | 53.3 |
| MetaMath-LB-15B Yoon et al. (2024) | 93.2 | 50.0 | 63.6 | 67.6 | 63.2 | 61.6 | 42.0 | 60.0 | 41.6 | 36.4 | 52.8 | 48.0 | 53.5 |
| MetaMath-LB-20B Yoon et al. (2024) | 93.1 | 52.8 | 64.0 | 66.4 | 60.4 | 64.0 | 45.2 | 58.8 | 49.2 | 47.2 | 53.6 | 52.4 | 55.8 |
| R1-Distill-Qwen-14B Guo et al. (2025) | 1030.7 | 66.0 | 77.2 | 83.6 | 80.4 | 74.4 | 78.4 | 82.4 | 22.4 | 22.4 | 74.8 | 79.6 | 67.4 |
| R1-Distill-Qwen-32B Guo et al. (2025) | 1353.8 | 77.6 | 82.8 | 85.2 | 85.6 | 79.6 | 83.2 | 84.8 | 14.4 | 38.0 | 82.4 | 85.6 | 72.7 |
| Gemma-3-12B-it Team et al. (2025) | 238.2 | 55.6 | 74.4 | 83.2 | 81.2 | 64.8 | 74.0 | 74.8 | **71.2** | **73.2** | 78.4 | 79.2 | 73.6 |
| Gemma-3-27B-it Team et al. (2025) | 461.7 | 64.8 | 83.2 | 88.4 | 84.0 | 72.4 | 79.2 | 83.2 | **78.0** | **76.0** | 84.4 | 84.4 | 79.8 |
| Qwen3-14B Team et al. (2025) | 1575.2 | **85.2** | 83.6 | **94.8** | **88.4** | **87.2** | 77.6 | **94.0** | **63.6** | **80.0** | **86.8** | 85.2 | **84.2** |

because s1 models are already generating extensive reasoning chains so further lengthening the CoTs have minimal benefits. Similar trends of language-specific improvements are also observed in other s1 model sizes (Appendix B).

**Comparison against state-of-the-art models.** Table 1 shows that crosslingual test-time scaling of s1 models can serve as a strong multilingual baseline for MGSM, as it outperforms all prior state-of-the-art models that involve finetuning on multilingual data such as MetaMath, MAPO and SLAM (Yu et al., 2024; She et al., 2024; Fan et al., 2025; Yoon et al., 2024). We believe this is because these prior studies use Llama as their base models, which generate significantly shorter reasoning traces and lack sophisticated reasoning behaviors such as verification and backtracking compared to Qwen models (Gandhi et al., 2025).

Surprisingly, 14B-sized s1 can even outperform recent state-of-the-art reasoning models twice its size, namely Deepseek's R1-Distill-Qwen-32B (Guo et al., 2025) and Google's Gemma-3-27B-it (Team et al., 2025). We observe that R1-Distill-Qwen has substantially poorer performance on sw and te than its base model Qwen-Instruct (first row), suggesting that their 800k samples of English and Chinese training data (Guo et al., 2025) leads to *catastrophic forgetting* of lower-resource languages. In contrast, s1 is only trained with 1k English samples for only 5 epochs (Muennighoff et al., 2025), which leads to minimal forgetting and better crosslingual generalization. While the multilingual Gemma-3 models outperform s1 on low-resource languages, probably due to these languages being incorporated during reasoning finetuning, its performance gap against s1 on high-resource languages may be attributed to the shorter reasoning thinking time. Qwen3 (Qwen Team, 2025) is the most performant model due to its long reasoning capability and extensive multilingual training data.

## 4.3 CROSSLINGUAL TRANSFER ON DIFFERENT MODELS AND TRAINING LANGUAGES

**Different base models.** We experimented with a different base model, Llama3.1-8B-Instruct (Grattafiori et al., 2024), and follow the training procedure of Muennighoff et al. (2025) that produces s1 from Qwen base models. Figure 4 shows that the benefits of extended thinking time generalize beyond the Qwen family, with Llama3.1-8B-Instruct exhibiting consistent accuracy improvements across languages as thinking time increases on MT-AIME2024. The performance trends closely parallel those observed with Qwen models, confirming that our findings represent a broader phenomenon rather than a family-specific characteristic.

**Different high-resource reasoning language.** To verify that improvements generalize beyond English, we used translated s1 datasets from Barua et al. (2025) and retrained s1-14B models with Mandarin Chinese and French as the reasoning training language. Figure 5 shows that models trained on Chinese, English, and French achieve similar performance on 8 low-resource languages of MT-AIME2024 benchmark (cy, kn, ml, gu, ne, af, mr, sq), with all three curves closely

> **Box 1.** Example of *quote-and-think* language-mixing pattern demonstrated by s1 where non-English phrases are mentioned in quotation marks and used for reasoning.
>
> **Question**: ローブを作成するには、青色の繊維を2巻分、白色の繊維をその半分用いる必要があります。全体で何巻必要ですか？
> (To make a robe, you need 2 rolls of blue fiber and half of the white fiber. How many rolls are needed total?)
>
> **s1's Thinking**: Okay, let's see. ... The problem says "白色の繊維をその半分用いる" which is white fibers the half amount. "その" refers to the blue fibers, right? So half of the blue fibers, which is 2 rolls, so half is 1. So 2 + 1 is 3. Yeah, that seems right. Wait, sometimes these problems can be tricks, but I don't think so. If it had said "白色の繊維を2巻分とその半分" that would be different, but no. It's "青色の繊維を2巻分、白色の繊維をその半分". So white is half of blue. So calculation is straightforward. Blue 2, white 1, total 3. ...

overlapping as thinking time increases. This demonstrates that cross-lingual reasoning transfer is not English-specific–any high-resource language can effectively improve reasoning in low-resource languages.

## 5 LANGUAGE-MIXING BEHAVIORS

We notice that s1 models can mix languages in their CoT reasoning under test-time scaling. Given a lack of systematic study of language mixing behaviors in crosslingual reasoning in prior work (Guo et al., 2025; Ghosh et al., 2025), we analyze linguistic behaviors of s1 in multilingual math reasoning tasks to address RQ2. We report our results on all the ten non-English languages in MGSM benchmark, and we refer our readers to Appendix C.1 for detailed methodology.

**Dominant Language in Model Outputs.** We observe that, after reasoning finetuning on 1k English reasoning samples, s1 generates in English language at least 90% of the time (Figure 6). This model behavior is the *complete opposite* of its base model Qwen, which always generates in the same language as question even when prompted with in-context English CoT samples (Appendix C.3).

**Language-Mixing Patterns During Reasoning.** Figure 6 shows that, in the remaining cases when s1 mixes languages during reasoning, it primarily follows a sophisticated pattern to which we refer as **quote-and-think**. Particularly, s1 will first quote certain words or phrases, often from the input question, and then interpret their meanings and implications during its thinking process. This is demonstrated by the quoted phrase "白色の繊維をその半分用いる" and s1's literal translation "white fibers the half amount" in Box 1. In linguistics, this type

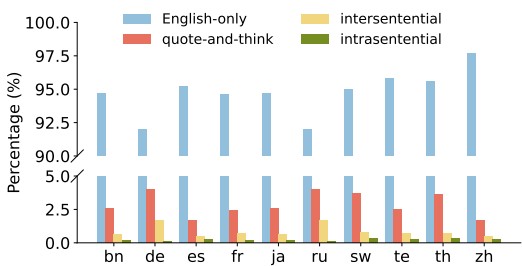

Figure 6: Breakdown of language-mixing patterns in s1's reasoning. Percentage indicates the probability of a sentence being English only, quoting non-English phrases (quote-and-think), entirely being in a different language (intersentential), or mixing different languages within the same sentence (intrasentential).

of language-mixing is known as foreign-language quotation (De Brabanter, 2004), and differs from the language confusion phenomenon exhibited by LLMs (Marchisio et al., 2024). This language-mixing behavior happens due to crosslingual generalization of the quoting-and-thinking reasoning characteristic in s1's English finetuning data, and we provide more quantitative analysis in Appendix C.4. We also measure the causal effects of the quote-and-think behavior in Appendix C.5.

We want to emphasize that the *quote-and-think pattern goes beyond simple translation*. As demonstrated in Box 1, s1 builds upon the extracted phrase and synthesizes a new multilingual setting where if the question had asked "白色の繊維を2巻分とその半分" (two and a half rolls of white fiber) it would have arrived at a different answer. Here, the model shows an understanding of how the syntactic structure in Japanese affects the semantic meaning of the math problem, which suggests that s1 is genuinely parsing and reasoning about the mathematical relationships expressed in Japanese and not merely translating the content to English before processing. This suggests that the multilingual capability of the base models is preserved for natural language understanding and allows s1 to reason about what it has understood about the question.

## 6 LANGUAGE FORCING

When a multilingual user interacts with LLMs, it is natural to expect the LLMs to respond in the language consistent with the user's query. Therefore, in RQ3, we are interested in understanding if we can perform *language forcing*——controlling an English-centric RLM to generate reasoning in a particular language—and if the difference in reasoning language affects performance.

### 6.1 METHODOLOGY: LANGUAGE FORCING TECHNIQUES

We experiment with the following language forcing techniques to control s1's reasoning language:

- **Translated Wait** (`translated_wait`): Building upon extrapolation budget forcing strategy that explicitly extends reasoning traces (Muennighoff et al., 2025), once the model finishes English reasoning, we append a translated "Wait" token as an intervention strategy to force the model to switch language and continue reasoning in our chosen language.
- **Prefix** (`prefix`): We append a prefix string translation-equivalent of "Okay, let me try to figure this out." at the beginning of the reasoning generation in order to guide the model's generation in our chosen language. We also apply the `translated_wait` strategy and append the translated "Wait" token.
- **System Prompt** (`system`): We use a system prompt to control the language use in model generation. Specifically, we translate the system prompt "You are a helpful assistant."[1] into our chosen language and add the translation-equivalent of the instruction "You must think and answer only in {language}".
- **Combined** (`combined`): This method uses all the techniques above to maximize control over the model's reasoning language.

Our results on MGSM benchmark show that we need a combination of all techniques, which is the **combined** method, to achieve nearly 100% success rate in forcing s1 to think in our specified reasoning language. Due to space constraint, we refer our readers to Appendix D.3 to see the full result comparison of different language forcing techniques.

### 6.2 CROSSLINGUAL LANGUAGE FORCING RESULTS

We explore if there is a particular language that is best served as reasoning language for s1 on the MGSM benchmark (which contains $M = 11$ different languages). Particular, for each query language $m \in M$, we force the model to reason in all $M$ possible languages using the `combined` technique, resulting in an exhaustive $M \times M$ query-reasoning language-pair analysis.[2]

**Performance comparison of reasoning languages.** Table 2 shows that reasoning in HRLs such as en, fr, or de yield similarly high performance (the accuracy difference is within 1 to 2 points), with English being the most performant reasoning language and French being the close second. We discover two surprising findings: first, even though the Qwen2.5 base model is highly pretrained in Chinese (Yang et al., 2024), it is not necessarily the best reasoning language, even when the question is asked in zh; second, neither reasoning in en nor in query language necessarily yields the

---

[1] We remove the part of "You are Qwen, created by Alibaba Cloud." because English proper nouns like 'Qwen' and 'Alibaba' do not have translation equivalents in many non-English languages.

[2] This analysis is is computationally heavy, so we only focus on 14B-sized s1 models.

Table 2: Performance scores across different reasoning languages given query language. We use 11 color codes to *rank each row* to highlight the high- (blue) and low-performing (red) reasoning language given a query language. We also **bold** the best-performing reasoning language. Lastly, we use ↘ to indicate the average accuracy when the reasoning language is the same as query language (i.e., average of the diagonals).

| Query Language | Reasoning Language | | | | | | | | | | | Range (max - min) |
|---|---|---|---|---|---|---|---|---|---|---|---|---|
| | bn | de | en | es | fr | ja | ru | sw | te | th | zh | |
| bn | 79.2 | 85.2 | **86.8** | 84.4 | 81.6 | 81.2 | 83.6 | 62.4 | 75.6 | 80.8 | 81.2 | 24.4 |
| de | 88.4 | 89.2 | 90.4 | 88.8 | **90.8** | 90.0 | 87.6 | 75.6 | 78.4 | 88.0 | 89.6 | 15.2 |
| en | 93.2 | 94.4 | 94.4 | 95.2 | 94.8 | 94.4 | 93.2 | 84.0 | 84.0 | 94.8 | **96.8** | 12.8 |
| es | 86.4 | 92.4 | **93.6** | 93.6 | 92.4 | 90.8 | 93.2 | 76.6 | 82.8 | 90.0 | 90.8 | 17.0 |
| fr | 87.2 | 87.2 | 88.4 | 87.2 | 88.0 | **89.6** | 88.4 | 72.8 | 77.6 | 87.2 | 88.0 | 16.8 |
| ja | 79.2 | 84.8 | 83.6 | 81.6 | **85.6** | 82.0 | 84.8 | 71.6 | 74.0 | **85.6** | 83.6 | 14.0 |
| ru | 89.2 | 91.2 | 92.4 | 89.6 | **93.6** | 92.0 | 92.4 | 77.6 | 80.8 | 90.0 | 91.2 | 16.0 |
| sw | 45.6 | 58.8 | **59.6** | 55.2 | 55.6 | 47.6 | 48.4 | 44.4 | 32.4 | 45.2 | 52.0 | 27.2 |
| te | 53.2 | 56.4 | **60.0** | 56.4 | **60.0** | 57.2 | 55.2 | 34.8 | 54.4 | 53.6 | 52.8 | 25.2 |
| th | 80.8 | 88.4 | 89.2 | 88.4 | **91.2** | 87.2 | 87.2 | 66.4 | 69.2 | 86.4 | 88.8 | 24.8 |
| zh | 85.2 | 86.8 | 89.6 | 87.2 | 86.8 | 88.8 | **90.8** | 73.6 | 77.2 | 86.0 | 89.2 | 17.2 |
| AVG | 78.9 | 83.2 | **84.4** | 82.6 | 83.7 | 81.9 | 82.3 | 67.3 | 71.5 | 80.7 | 82.2 | ↘81.2 |

best performance—quite the contrary, even reasoning in languages that are usually less represented in pretraining data (Joshi et al., 2020) such as ru and th can achieve the best performance for query languages in other families such as ja. Lastly, we observe that languages that are considered as slightly less-resourced (Joshi et al., 2020) such as th and bn still achieve nearly 80% overall accuracy, but further lower-resourced languages such as sw or te result in substantially lower overall accuracy. Our results are consistent with the findings by concurrent work (Qi et al., 2025).

**Choice of query language.** Table 2 sheds light on whether we should translate inputs into HRLs such as English for reasoning tasks, which has proven to be an effective strategy (Qin et al., 2023; Zhu et al., 2024). Our results are consistent with prior work: merely translating the question from Swahili to German can boost the accuracy from 59.6 to 90.8 even when the model reasons in French—a language that s1 is not trained to reason in. Besides, based on the range column, which measures difference between the best and worst reasoning languages for a particular query language, the model is less sensitive to query language in HRLs than in LRLs as exhibited by the smaller range. In other words, querying s1 in HRLs increases the model's consistency in achieving the same correct answer with different reasoning languages.

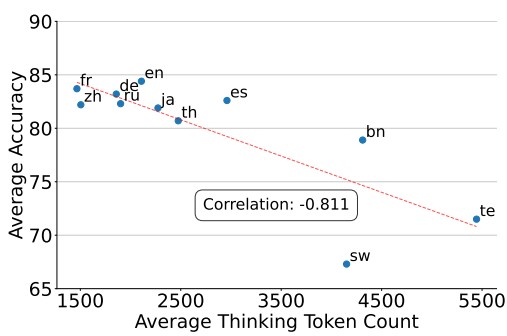

Figure 7: MGSM accuracy against number of thinking tokens in s1 models' outputs in different reasoning languages.

**Inference cost analysis.** Our analysis of inference costs across reasoning languages in Figure 7 reveals a significant negative correlation (-0.811) between token count and mathematical problem-solving accuracy. Reasoning in LRLs not only underperform their HRL counterpart (with accuracy below 80%), but they also demand substantially more computational resources at test-time. For instance, reasoning in Swahili requires approximately 3.5 times more compute than French for the same tasks. One potential reason for this is *tokenization inefficiency*: common terms in CoT like "but" is tokenized as 5 in Swahili ("hata hivyo") but only 2 subtokens in French ("mais"), and "for example" 4 in Swahili ("kwa mfano") but only 2 in French ("par exemple"). These additional tokens will accumulate throughout reasoning chains and lead to larger inference costs.

# 7 DISCUSSION, LIMITATIONS AND FUTURE WORK

**Crosslingual generalization from English finetuning.** Our findings from `RQ1` contrast the conclusion drawn by recent work (Son et al., 2025) that "test-time scaling may not generalize as effectively to multilingual tasks." Our results on different benchmarks of varying difficulty levels and languages suggest that the limitation observed by Son et al. (2025) is due to their usage of 1.5B models. Furthermore, Son et al. (2025) posit that multilingual generalization of test-time scaling would occur for "significantly larger" models with at least 70B parameters, but we discover a substantially smaller parameter threshold at 3B parameters, above of which RLMs will benefit from English finetuning on multilingual tasks.

**Preservation of multilingual generation capability.** One notable finding from `RQ2` and `RQ3` is that s1 remains capable of generating text in different languages and experiences minimal catastrophic forgetting—a phenomenon where the model loses its ability to generate fluent text in other languages after language-specific supervised finetuning (Yong et al., 2023; Kotha et al., 2024). In contrast, R1-Distill-Qwen baseline experiences significant catastrophic forgetting for low-resource languages. This suggests that data-efficient finetuning with a small number of reasoning finetuning steps (s1 is only trained with 1k English samples for 5 epochs, but R1-Distill models are trained with 800k samples) is advisable for English-centric reasoning finetuning to preserve multilingual capability.

**Training RLMs with multilingual data.** One potential solution to mitigate poorer reasoning in low-resource languages would be to curate multilingual reasoning training data with wide language and domain coverage. Our work only focuses on English reasoning training, and we leave the exploration of multilingual reasoning training for future work. Future work should compare the effectiveness of different multilingual augmentation techniques such as back-translation (Edunov et al., 2018) or synthetic data generation (Whitehouse et al., 2023; Yong et al., 2024) for reasoning tasks.

**Reasoning in low-resource languages (LRLs).** Reasoning in LRLs can be challenging in deployment due to the the higher inference costs for test-time scaling of RLMs. We believe that one solution is to mitigate the unfairness in tokenization for LRLs (Petrov et al., 2023). Future work should focus on developing more equitable tokenization strategies for reasoning across diverse languages (Liang et al., 2023; Han et al., 2025; Xue et al., 2022).

**Conclusion and limitations.** We show that scaling up thinking tokens of English-centric reasoning language models can improve multilingual math reasoning performance as the model performs "quote-and-think" language-mixing pattern. We also show that English-centric RLMs reason poorly in low-resource languages contexts, thus highlighting the need for future work to develop more inclusive multilingual reasoning approaches that can better serve diverse linguistic communities. A systematic expert study on naturalness of the generated CoTs would provide insights on how to curate data and improve models' reasoning in low-resource languages. Our analysis focuses only on sequential test-time scaling paradigm and one common type of RLMs, such as s1, which are created from supervised finetuning on distilled reasoning data. Future work can expand our analysis to study multilingual generalization of reasoning reinforcement learning.

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

APPENDIX

## A  LLM USAGE

Our work used AI Assistants such as ChatGPT and Grammarly for spell-checking and fixing minor grammatical mistakes. We also use Claude Code to write parts of our codebase.

## B  FURTHER DETAILS ON CROSSLINGUAL TEST-TIME SCALING

### B.1  MGSM

Table 3, Table 4, Table 5, and Table 6 shows the s1 performance against Qwen baselines on the MGSM benchmark. Relative accuracy difference measures the relative gains for s1 under extrapolation budget forcing compared to zero-shot prompting Qwen models (first row), except for Table 6 where the relative accuracy is measured for s1 under truncation budget forcing.

Table 3: MGSM performance comparison against 32B-sized s1 model with maximum 8k thinking tokens.

| Models | avg len | bn | de | en | es | fr | ja | ru | sw | te | th | zh | AVG |
|---|---|---|---|---|---|---|---|---|---|---|---|---|---|
| Qwen-32B-Instruct | 365.6 | 82.0 | 79.6 | 84.0 | 80.0 | 69.2 | 80.8 | 77.2 | 54.0 | 56.4 | 84.4 | 84.0 | 75.6 |
| + 8-Shot EN-CoT | 264.7 | 82.0 | 80.4 | 89.6 | 84.8 | 66.8 | **85.2** | 77.6 | 56.8 | 55.6 | 84.8 | 84.8 | 77.1 |
| + 8-Shot Native-CoT | 179.9 | 82.4 | 78.0 | 90.0 | 87.2 | 66.8 | 80.8 | 74.0 | 57.2 | 59.6 | 87.2 | 88.0 | 77.4 |
| s1-32B (truncation) | 1682.1 | **89.2** | 86.8 | **94.8** | 90.0 | 84.8 | 84.8 | **91.2** | 66.8 | 65.6 | 87.2 | **88.8** | 84.5 |
| **s1-32B (extrapolation)** | 2610.1 | **89.2** | **88.8** | 93.2 | 89.6 | 83.6 | 84.8 | 88.8 | **70.8** | **68.4** | 88.0 | 88.0 | **84.8** |
| Relative accuracy difference (%) | | +8.8% | +11.6% | +11.0% | +12.0% | +20.8% | +5.0% | +15.0% | +31.1% | +21.3% | +4.3% | +4.8% | +12.2% |

Table 4: MGSM performance comparison against 7B-sized s1 model with maximum 8k thinking tokens.

| Models | avg len | bn | de | en | es | fr | ja | ru | sw | te | th | zh | AVG |
|---|---|---|---|---|---|---|---|---|---|---|---|---|---|
| Qwen-7B-Instruct | 537.7 | 59.2 | 69.2 | 78.0 | 72.8 | 66.4 | 67.2 | 71.2 | 13.6 | 33.2 | 68.8 | 79.6 | 61.7 |
| + 8-Shot EN-CoT | 537.8 | 62.0 | 72.4 | 86.4 | 72.4 | 62.4 | 68.0 | 76.8 | 15.2 | 33.2 | 70.4 | 76.8 | 63.9 |
| + 8-Shot Native-CoT | 480.1 | 65.2 | 74.4 | **90.4** | 76.4 | 65.2 | 71.6 | 68.8 | 18.4 | 20.8 | 69.6 | 76.8 | 63.4 |
| **s1-7B (truncation)** | 3767.1 | 65.2 | 82.8 | 88.8 | **86.0** | 82.0 | **78.8** | **86.4** | **21.6** | **38.8** | **80.0** | **83.6** | **72.2** |
| s1-7B (extrapolation) | 4363.5 | **70.8** | **84.0** | **90.4** | 83.6 | **84.4** | 74.8 | 84.4 | 19.2 | 36.4 | 78.4 | 82.8 | 71.7 |
| Relative accuracy difference (%) | | +19.6% | +21.4% | +15.9% | +14.8% | +27.1% | +11.3% | +18.5% | +41.2% | +9.6% | +14.0% | +4.0% | +16.2% |

Table 5: MGSM performance comparison against 3B-sized s1 model with maximum 8k thinking tokens.

| Models | avg len | bn | de | en | es | fr | ja | ru | sw | te | th | zh | AVG |
|---|---|---|---|---|---|---|---|---|---|---|---|---|---|
| Qwen-3B-Instruct | 1023.3 | 37.6 | 58.8 | 74.0 | 66.0 | 54.4 | 54.8 | 64.8 | 9.2 | 7.6 | 56.8 | 68.4 | 50.2 |
| + 8-Shot EN-CoT | 281.3 | 48.0 | **67.6** | 79.2 | 71.2 | 65.2 | 58.0 | 70.4 | **12.4** | 14.8 | 60.4 | 68.4 | 56.0 |
| + 8-Shot Native-CoT | 1657.0 | 36.0 | 63.2 | 80.0 | 70.8 | 58.4 | 52.0 | 62.0 | 9.6 | 9.6 | 59.2 | 70.4 | 51.9 |
| s1-3B (truncation) | 4813.3 | **56.8** | 66.8 | **82.0** | 74.4 | 69.6 | 60.4 | 72.4 | 10.4 | 16.8 | **68.8** | 72.0 | 59.1 |
| **s1-3B (extrapolation)** | 5367.1 | 55.2 | 65.6 | 81.6 | **76.4** | **71.6** | 60.8 | **74.4** | 9.6 | **20.0** | 68.8 | 74.0 | **59.8** |
| Relative accuracy difference (%) | | +46.8% | +11.6% | +10.3% | +15.8% | +31.6% | +10.9% | +14.8% | +4.3% | +163.2% | +21.1% | +8.2% | +19.1% |

### B.2  PERFORMANCE COMPARISON ON POLYMATH

Table 7 compares s1-32B model against other non-reasoning and reasoning models on the high-difficulty PolyMath dataset, which is the difficulty level that s1 is trained on). The s1-32B model substantially outperforms all non-reasoning language models, including much larger variants. This validates the core premise that English reasoning finetuning can effectively generalize to complex mathematical problems in diverse linguistic contexts.

We observe that s1-32B underperforms other RLMs, which is likely due to model capacity constraints. The competing RLMs often operate at significantly larger scales (such as Deepseek-R1-671B) or with more extensive reasoning training (such as Qwen-QwQ-32B and Qwen-3-225B-A22B-Thinking), making the capacity disparity the most plausible explanation for the reasoning performance differences.

Table 6: MGSM performance comparison against 1.5B-sized s1 model with maximum 8k thinking tokens. We didn't run extrapolation budget forcing since without it, s1 already generates extremely long CoTs.

| Models | avg len | bn | de | en | es | fr | ja | ru | sw | te | th | zh | AVG |
|---|---|---|---|---|---|---|---|---|---|---|---|---|---|
| Qwen-1.5B-Instruct | 2991.7 | 10.4 | 35.6 | 66.0 | 52.8 | 41.2 | 31.2 | 43.6 | **2.0** | 1.2 | 31.2 | 56.0 | 33.7 |
| + 8-Shot EN-CoT | 1100.3 | 21.6 | 46.4 | **70.0** | 58.0 | 55.2 | 37.2 | 51.6 | **2.8** | **6.4** | 41.2 | 52.0 | 40.2 |
| + 8-Shot Native-CoT | 1729.9 | 14.0 | 44.4 | **71.6** | 52.4 | 41.6 | 34.4 | 39.2 | **3.2** | 1.6 | 33.2 | 54.0 | 35.4 |
| s1-1.5B (truncation) | 8227.2 | **27.6** | **51.2** | 66.8 | **62.8** | **56.0** | **43.2** | **55.6** | 1.6 | **6.4** | **46.4** | **58.8** | **43.3** |
| Relative accuracy difference (%) | | +165.4% | +43.8% | +1.2% | +18.9% | +35.9% | +38.5% | +27.5% | -20.0% | +433.3% | +48.7% | +5.0% | +28.5% |

Table 7: The accuracy for non-reasoning and reasoning language models for PolyMath (Wang et al., 2025b) of high difficulty level. Models with "†" are closed-source, and the numbers are taken directly from the PolyMath paper (Wang et al., 2025b).

| | en | zh | ar | bn | de | es | fr | id | it | ja | ko | ms | pt | ru | sw | te | th | vi | *avg.* |
|---|---|---|---|---|---|---|---|---|---|---|---|---|---|---|---|---|---|---|---|
| **Non-Reasoning LLMs** | | | | | | | | | | | | | | | | | | | |
| Llama-3.3-70B-Instruct | 14.4 | 6.4 | 5.6 | 1.6 | 7.2 | 4.8 | 5.6 | 4.8 | 11.2 | 5.6 | 4.8 | 11.2 | 10.4 | 10.4 | 6.4 | 4.8 | 5.6 | 10.4 | 7.3 |
| Qwen-2.5-72B-Instruct | 14.4 | 12.0 | 11.2 | 10.4 | 12.0 | 12.8 | 12.0 | 9.6 | 11.2 | 11.2 | 10.4 | 13.6 | 11.2 | 13.6 | 5.6 | 12.0 | 14.4 | 11.2 | 11.6 |
| Qwen-2.5-Math-72B-Instruct | 16.0 | 18.4 | 17.6 | 16.8 | 22.4 | 19.2 | 16.8 | 13.6 | 13.6 | 16.8 | 17.6 | 16.8 | 19.2 | 16.0 | 15.2 | 16.0 | 12.8 | 16.8 | 16.8 |
| Deepseek-v3 | 16.8 | 17.6 | 16.8 | 15.2 | 17.6 | 16.8 | 16.8 | 12.8 | 16.0 | 16.8 | 17.6 | 17.6 | 20.0 | 19.2 | 12.8 | 12.8 | 18.4 | 16.0 | 16.5 |
| Qwen-2.5-Max† | 12.0 | 17.6 | 16.8 | 10.4 | 13.6 | 16.8 | 16.0 | 16.0 | 14.4 | 10.4 | 14.4 | 16.8 | 16.0 | 16.0 | 14.4 | 13.6 | 12.8 | 15.2 | 14.6 |
| Claude-3.7-sonnet† | 21.6 | 17.6 | 14.4 | 13.6 | 17.6 | 15.2 | 12.8 | 16.8 | 12.0 | 12.8 | 16.0 | 13.6 | 15.2 | 14.4 | 12.8 | 16.0 | 15.2 | | 15.0 |
| ChatGPT-4o-latest† | 22.4 | 21.6 | 20.0 | 16.8 | 23.2 | 20.0 | 24.8 | 16.0 | 20.0 | 22.4 | 20.0 | 23.2 | 20.8 | 21.6 | 17.6 | 17.6 | 17.6 | 20.8 | 20.4 |
| GPT-4.5-preview† | 34.4 | 25.6 | 24.8 | 24.0 | 24.8 | 29.6 | 27.2 | 27.2 | 28.8 | 27.2 | 25.6 | 29.6 | 27.2 | 31.4 | 25.6 | 24.0 | 26.4 | 29.6 | 27.4 |
| **Reasoning LLMs** | | | | | | | | | | | | | | | | | | | |
| Deepseek-R1-671B | 48.8 | 46.4 | 50.4 | 46.4 | 52.8 | 55.2 | 52.8 | 52.0 | 56.8 | 51.2 | 46.4 | 51.2 | 52.0 | 51.2 | 51.2 | 44.8 | 49.6 | 52.8 | 50.7 |
| Qwen-QwQ-32B | 62.4 | 55.2 | 47.2 | 50.4 | 63.2 | 60.0 | 58.4 | 56.0 | 56.8 | 44.8 | 47.2 | 57.6 | 59.2 | 55.2 | 45.6 | 43.2 | 47.2 | 60.0 | 53.9 |
| Qwen-3-235B-A22B-Thinking | **66.4** | 62.9 | **62.4** | **62.4** | 63.2 | **64.8** | **66.4** | 60.8 | **70.4** | 61.6 | **64.8** | 59.2 | **64.8** | 60.0 | **60.0** | **60.0** | **64.0** | **65.6** | **63.3** |
| Claude-3.7-sonnet-thinking† | 36.0 | 38.4 | 36.8 | 38.4 | 35.2 | 29.6 | 32.0 | 36.8 | 38.4 | 34.4 | 37.6 | 39.2 | 37.6 | 40.8 | 40.0 | 38.4 | 37.6 | 33.6 | 36.7 |
| Gemini-2.0-flash-thinking† | 43.2 | 43.2 | 42.4 | 44.0 | 40.8 | 42.4 | 48.0 | 41.6 | 44.0 | 36.8 | 40.8 | 44.0 | 46.4 | 47.2 | 41.6 | 36.8 | 46.4 | 43.2 | 42.9 |
| Gemini-2.5-pro† | **66.4** | **66.4** | **62.4** | **62.4** | **65.6** | **64.8** | 63.2 | 59.2 | 68.8 | **64.8** | 61.6 | **60.8** | 63.2 | **62.4** | 56.8 | **60.0** | 50.4 | 60.0 | 62.2 |
| OpenAI-o1-mini† | 46.4 | 44.8 | 37.6 | 37.6 | 36.0 | 43.2 | 40.0 | 40.8 | 40.8 | 41.6 | 43.2 | 38.4 | 40.8 | 38.4 | 36.0 | 40.0 | 42.4 | 41.6 | 40.5 |
| OpenAI-o3-mini-medium† | 54.4 | 52.8 | 51.2 | 52.8 | 53.6 | 51.2 | 50.4 | 56.0 | 45.6 | 52.0 | 50.4 | 50.4 | 50.4 | 39.2 | 51.2 | 41.6 | 44.8 | 52.8 | 50.0 |
| **s1-32B (8000 tokens)** | 42.1 | 34.1 | 28.5 | 23.3 | 27.6 | 30.7 | 26.8 | 24.5 | 35.1 | 36.2 | 28.8 | 25.9 | 33.1 | 35.4 | 20.6 | 20.0 | 22.9 | 26.3 | 28.2 |

## B.3 PERFORMANCE BREAKDOWN ANALYSIS: LANGUAGE FAMILY AND DATA DISTRIBUTION

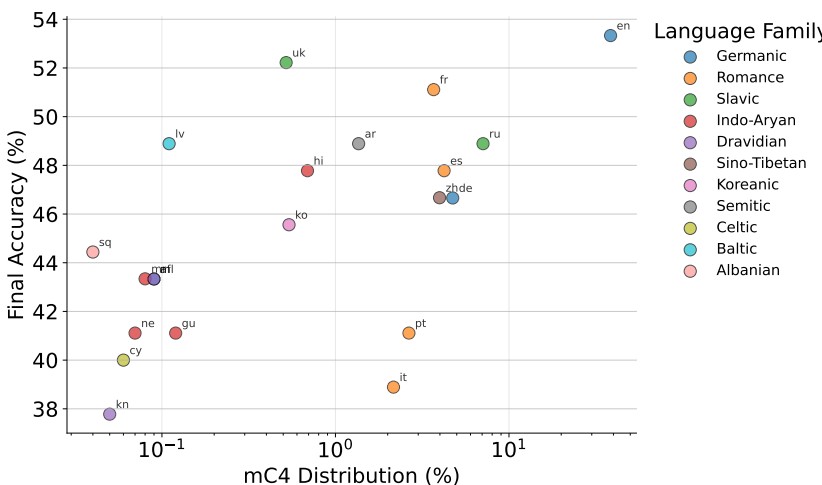

Figure 8: Performance analysis of Figure 2 based on mC4 distribution and language family.

**Language Family Analysis.** Figure 8 presents a scatter plot analyzing final accuracy across different language families and their representation in the mC4 dataset. First, the plot reveals that languages from the same family (indicated by color) do not cluster together in terms of performance. For instance, within the Romance family, we observe substantial variation with Italian (it) achieving 51% accuracy while Portuguese (pt) reaches only 41%. Similarly, Slavic languages show diverse performance ranging from Russian (ru) at 49% to Ukrainian (uk) at 44%. Second, the six top-

performing languages (English, Ukranian, French, Russian, Arabic, Latvian) span different language families, further demonstrating that language family membership is not a primary determinant of model performance. This pattern suggests that linguistic typology alone does not explain the observed performance variations across languages.

**Data Distribution.** When examining the relationship between mC4 data distribution (x-axis) and final accuracy, we observe a stronger positive correlation (Pearson correlation coefficient = 0.54). This indicates that languages with greater representation in the continued pretraining data tend to achieve higher performance. This supports our hypothesis that the base model's language representation in pretraining is the dominant factor determining crosslingual reasoning transfer performance. However, the mC4 distribution may not accurately reflect Qwen's actual pretraining corpus, which is proprietary and might differ substantially, particularly given Qwen's focus on Chinese and other specific languages.

## C  FURTHER DETAILS ON LANGUAGE-MIXING BEHAVIORS

### C.1  METHODOLOGY

To filter out language-mixed sentences, we first identify the *dominant language*, also known as matrix language, of the generated response using the state-of-the-art language identification library `lingua`. Then, we use the NLP library `stanza` to perform sentence segmentation according to the matrix language and obtain individual sentences. Finally, we use `lingua` to annotate the language label of each sentence and of each individual word token in the sentence.

We classify language-mixing patterns into three categories: (1) quote-and-think, where words or phrases in foreign language are quoted in quotation marks; (2) intersentential, where the entire sentence is in a language entirely different from generation dominant language, and (3) intrasentential, where words, phrases or clauses of different languages are present in the same sentence. We refer our readers to Appendix C.2 for our annotation procedures.

### C.2  ANNOTATION FOR LANGUAGE-MIXING PATTERNS IN S1 REASONING

We collect the language label with the highest probability assigned to *the entire sentence*, and we label a sentence belonging to "intersentential" language-mixing for s1 if the sentence is non-English, as the dominant language of s1's overall output is English. We then check language labels for individual word tokens. If there are mixing of different languages *within the same sentence*, and quotation marks are present around the non-English words or phrases, then the sentence is assigned with the "quote-and-think" label. Otherwise, if quotation marks are not present, the sentence is assigned with the "intrasentential" label.

### C.3  DOMINANT LANGUAGE IN 14B-SIZED MODEL OUTPUTS

Figure 9 shows the dominant language distribution in model outputs when MGSM questions are asked in Japanese (ja), Russian (ru), Thai (th), and Mandarin Chinese (zh).

### C.4  QUOTE-AND-THINK PATTERN IN S1'S TRAINING DATA

Among 1k English training samples of s1 models, 68.3% of the samples exhibit the quote-and-think pattern, among which at least half of them involves directly copying from the question prompts. This suggests that the quote-and-think language-mixing pattern is due to crosslingual transfer of the original s1 model's learned behavior of quoting phrases from question prompts during its long CoTs thinking process.

### C.5  CAUSAL EFFECTS OF QUOTE-AND-THINK PATTERN

To evaluate if quote-and-think pattern is causal, we measure the accuracy drop when we remove quote-and-think phrases. Specifically, we replaced the phrases within the quotation mark with an empty string and force s1 to regenerate CoT from the end of the quotation mark. We repeated this for 5 different runs and report the average accuracy. Table 8 shows a decrease in MGSM accuracy

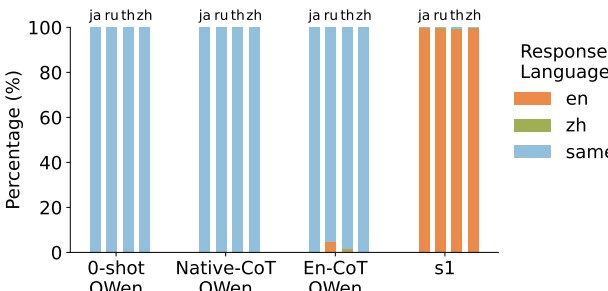

Figure 9: Proportion of dominant languages used by 14B-sized models' responses when queried with Japanese (ja), Russian (ru), Thai (th), and Mandarin Chinese (zh) languages. "same" indicates that the response language is the same as query language.

| | bn | de | en | es | fr | ja |
|---|---|---|---|---|---|---|
| Before removal | $80.1 \pm 1.2$ | $84.2 \pm 0.9$ | $90.2 \pm 0.7$ | $81.7 \pm 1.1$ | $79.9 \pm 1.3$ | $85.4 \pm 0.8$ |
| After removal | $76.3 \pm 1.3$ | $83.4 \pm 1.1$ | $88.4 \pm 0.8$ | $80.8 \pm 1.2$ | $78.4 \pm 1.4$ | $82.3 \pm 1.0$ |

on those instances that have quote-and-think behaviors, with the largest drop for lower-resource languages such as Swahili (sw) and Telugu (te).

### C.6 Fine-Grained Analysis of s1-32B's Intrasentential Language Mixing

We perform human annotations on the intrasententially language-mixed sentences during reasoning and classify if each sentence belongs to one of the following categories: (1) **extract-and-explain**, where the non-English phrases are taken directly from the original input prompt but *without* quotation marks given (this resembles quote-and-think but no quotation marks are generated around the non-English phrases); (2) **insertional code-switching**, where non-English lexical items (usually nouns) are inserted into the morphosyntactic frame of the English sentence (an example would be "I want to eat *nasi goreng*" where *nasi goreng* is a Malay word for fried rice), and (3) **clause-level code-switching**, where switching between two languages within a single sentence structure at clausal level (an example would be "I want to go to the library *dan bersedia untuk peperiksaan*" where *dan bersedia untuk peperiksaan* is a Malay clause for "and prepare for the exam".) Figure 10 demonstrates the distribution of each category, with extract-and-explain being the dominant language-mixing patterns.

## D Further Details on Language Forcing

### D.1 Example of CoTs after Language Forcing

Box 2 show the example of CoTs generated by s1 from the `combined` language-forcing strategy. This example contrasts Box 1 where Box 1 showcases s1's natural CoTs without language forcing.

Note that we still observe the sophisticated *quote-and-think* pattern where here, English phrases are quoted for reasoning (the dominant language is Japanese). Specifically, in that sentence where quote-and-think occurs, it is translated to 'if this question was in English, it would be "how many rolls are needed in total?"'

| | ru | sw | te | th | zh |
|---|---|---|---|---|---|
| Before removal | $83.7 \pm 1.0$ | $53.8 \pm 2.8$ | $59.1 \pm 2.4$ | $80.9 \pm 1.1$ | $85.4 \pm 0.9$ |
| After removal | $80.5 \pm 1.2$ | $47.7 \pm 3.1$ | $53.6 \pm 2.5$ | $78.1 \pm 1.3$ | $82.3 \pm 1.1$ |

Table 8: Causal effects of quote-and-think phrases.

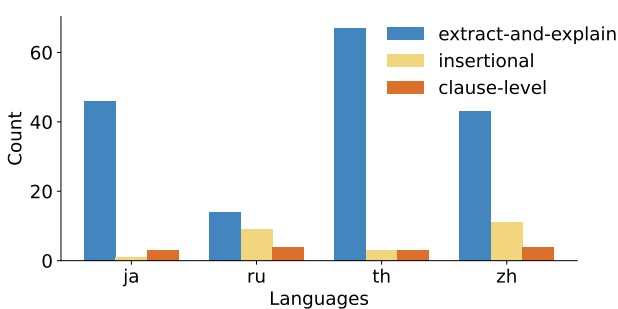

Figure 10: Breakdown of s1-32B's finegrained intrasentential language mixing patterns.

---

**Box 2. Example of CoTs prdocued from** `combined` **language forcing strategy.**

**Question**: ローブを作成するには、青色の繊維を2巻分、白色の繊維をその半分用いる必要があります。全体で何巻必要ですか？
(To make a robe, you need 2 rolls of blue fiber and half of the white fiber. How many rolls are needed total?)

**s1's Thinking**: よし、解いてみよう。問題はローブを作るために必要な繊維の巻数を計算するものだね。青色の繊維が2巻必要で、白色の繊維はその半分、つまり1巻必要だ。だから合計で3巻か。でも待って、問題文をもう一度確認しよう。

「ローブを作成するには、青色の繊維を2巻分、白色の繊維をその半分用いる必要があります。全体で何巻必要ですか？」そうだ、青色が2巻、白色がその半分。つまり青色の半分は1巻だから、白色は1巻必要。だから合計3巻。これでいいのかな。でも、もしかして青色の2巻と白色の半分を足すと、2 + 0.5 = 2.5巻って意味じゃないのだろうか。でも問題の言い回しが少しややこしいな。

待って、原文をもう一度。「青色の繊維を2巻分、白色の繊維をその半分用いる」というのは、青色が2巻、そして白色が「その半分」。つまり、「青色の半分」なのか、それとも「2巻の半分」なのか。日本語の文法で考えると、「その半分」は直前の数詞、「2巻分」を指している可能性が高い。つまり、青色2巻、白色は2巻の半分なので1巻。だから合計3巻。でも、もし「青色の繊維を2巻分、白色の繊維をその半分」というのが、青色が2巻、白色が青色の半分、つまり1巻、という意味なら、同じく3巻。結局同じ結果になるのか。

それに、問題が「全体で何巻必要ですか？」と聞いているので、必要な総巻数を聞かれているのだろう。つまり、青色2巻、白色1巻、合計3巻。これが答えだと思われる。でも、もしこの問題が英語だと、"how many rolls are needed in total?"だと仮定すると、3 rolls が答え。でも、もしかして青色2巻、白色半分、つまり2.5巻という解釈もある。でも、問題文の日本語の表現からすると、青色2巻、白色1巻、合計3巻が妥当だと思う。
...

## D.2 LANGUAGE COMPLIANCE

Detailed results for language compliance can be found in Tables 9, 10 and 11 for 32B, 14B and 7B, respectively.

## D.3 IN-LANGUAGE BUDGET FORCING REASONING

Detailed results for in-language budget forcing reasoning performance on MGSM are presented in Table 12.

Table 9: Results for **s1-32B**.

| Strategy | Reasoning Language | avg len | Language Distribution | Language Compliance |
|---|---|---|---|---|
| translated_wait | bn | 3073.3 | ENGLISH: 99.6, BENGALI: 0.4 | 0.4 |
| | de | 2405.3 | ENGLISH: 51.6, GERMAN: 48.4 | 48.4 |
| | en | 1833.1 | ENGLISH: 100.0 | 100.0 |
| | es | 2401.5 | ENGLISH: 88.4, SPANISH: 11.6 | 11.6 |
| | fr | 2379.7 | ENGLISH: 90.8, FRENCH: 8.8, CHINESE: 0.4 | 8.8 |
| | ja | 2515.1 | ENGLISH: 70.8, JAPANESE: 29.2 | 29.2 |
| | ru | 2601.1 | ENGLISH: 90.4, RUSSIAN: 9.2, CHINESE: 0.4 | 9.2 |
| | sw | 2611.4 | ENGLISH: 100.0 | 0.0 |
| | te | 3821.3 | ENGLISH: 99.6, TELUGU: 0.4 | 0.4 |
| | th | 1894.7 | ENGLISH: 99.6, CHINESE: 0.4 | 0.0 |
| | zh | 1776.3 | ENGLISH: 94.4, CHINESE: 5.6 | 5.6 |
| prefix | bn | 3320.0 | BENGALI: 98.8, ENGLISH: 1.2 | 98.8 |
| | de | 1747.2 | GERMAN: 98.8, CHINESE: 1.2 | 98.8 |
| | en | 1729.7 | ENGLISH: 100.0 | 100.0 |
| | es | 2790.6 | SPANISH: 99.6, ENGLISH: 0.4 | 99.6 |
| | fr | 1822.9 | FRENCH: 100.0 | 100.0 |
| | ja | 2321.6 | JAPANESE: 98.8, ENGLISH: 1.2 | 98.8 |
| | ru | 1564.2 | RUSSIAN: 98.8, ENGLISH: 0.8, CHINESE: 0.4 | 98.8 |
| | sw | 3083.9 | SWAHILI: 91.2, ENGLISH: 8.4, JAPANESE: 0.4 | 91.2 |
| | te | 6912.7 | TELUGU: 95.2, ENGLISH: 4.8 | 95.2 |
| | th | 2126.3 | THAI: 77.6, CHINESE: 20.4, ENGLISH: 2.0 | 77.6 |
| | zh | 1150.1 | CHINESE: 99.6, ENGLISH: 0.4 | 99.6 |
| system | bn | 2693.8 | ENGLISH: 90.4, BENGALI: 9.6 | 9.6 |
| | de | 1979.3 | GERMAN: 100.0 | 100.0 |
| | en | 1728.3 | ENGLISH: 100.0 | 100.0 |
| | es | 2241.6 | ENGLISH: 69.6, SPANISH: 30.4 | 30.4 |
| | fr | 2346.0 | ENGLISH: 68.4, FRENCH: 31.2, CHINESE: 0.4 | 31.2 |
| | ja | 1805.7 | JAPANESE: 99.2, ENGLISH: 0.8 | 99.2 |
| | ru | 2180.0 | ENGLISH: 59.6, RUSSIAN: 39.2, CHINESE: 1.2 | 39.2 |
| | sw | 2721.2 | ENGLISH: 98.8, SWAHILI: 0.8, CHINESE: 0.4 | 0.8 |
| | te | 3869.3 | ENGLISH: 93.6, TELUGU: 6.4 | 6.4 |
| | th | 1930.3 | ENGLISH: 90.8, THAI: 8.0, CHINESE: 1.2 | 8.0 |
| | zh | 1162.1 | CHINESE: 100.0 | 100.0 |
| combined | bn | 3507.6 | BENGALI: 99.6, CHINESE: 0.4 | 99.6 |
| | de | 1845.2 | GERMAN: 100.0 | 100.0 |
| | en | 1582.6 | ENGLISH: 100.0 | 100.0 |
| | es | 2604.3 | SPANISH: 100.0 | 100.0 |
| | fr | 1726.5 | FRENCH: 99.2, CHINESE: 0.8 | 99.2 |
| | ja | 2127.4 | JAPANESE: 100.0 | 100.0 |
| | ru | 1523.6 | RUSSIAN: 98.8, ENGLISH: 0.4, GERMAN: 0.4, CHINESE: 0.4 | 98.8 |
| | sw | 3161.1 | SWAHILI: 98.4, ENGLISH: 1.6 | 98.4 |
| | te | 7036.0 | TELUGU: 97.6, ENGLISH: 2.4 | 97.6 |
| | th | 2043.3 | THAI: 90.8, CHINESE: 8.8, GERMAN: 0.4 | 90.8 |
| | zh | 1188.7 | CHINESE: 100.0 | 100.0 |

# E CROSS-DOMAIN GENERALIZATION

Since s1 models obtain strong crosslingual math performance with English-only training, a natural question to ask is whether such generalization extends to other non-math domains that may require knowledge recall or cultural reasoning. We address this research question using Global-MMLU (Singh et al., 2024), FORK (Palta & Rudinger, 2023), and COPAL-ID (Wibowo et al., 2024)

**In-domain (STEM) vs out-of-domain performance.** Figure 11 (a) shows that test-time scaling of thinking tokens (cyan line) substantially improves s1's performance on STEM subject domain. This strong crosslingual in-domain generalization[3] is consistent with our findings in Section 4. On the other hand, for out-of-domain subjects, we report *minimal cross-domain generalization* of test-time scaling from Figure 11 (a). Domains such as medicine do not benefit from scaling up thinking tokens, as increasing maximum thinking tokens from 0.5k to 4k tokens merely improves accuracy by only $+\Delta 0.8\%$ ($73.0\% \rightarrow 73.8\%$), and further scaling to 8000 thinking tokens even reduces accuracy by $-\Delta 2.0\%$ ($73.0\% \rightarrow 71.0\%$). Out of all non-STEM domains, business benefits the most from test-time scaling ($+\Delta 3.2\%$), but the accuracy gain still lags behind STEM domain ($+\Delta 11.5\%$) by a huge margin.

**Cultural-specific knowledge and reasoning.** For multilingual cultural benchmarks, we observe similar findings that there is minimal benefit of test-time scaling of s1. Figure 11 (b) shows that while reasoning finetuning improves overall model performance over Qwen baselines (dashed lines), scaling up test-time thinking compute does not improve performance. In fact, for the English FORK benchmark, increasing thinking tokens leads to substantially poorer performance. This is also known

---

[3] s1 training data includes OlympicArena dataset (Huang et al., 2024a) that encompasses various STEM subject knowledge such as biology and astronomy

Table 10: Results for **s1-14B**.

| Strategy | Reasoning Language | avg len | Language Distribution | Language Compliance |
|---|---|---|---|---|
| translated_wait | bn | 2457.1 | ENGLISH: 100.0 | 0.0 |
| | de | 1940.1 | ENGLISH: 76.4, GERMAN: 23.6 | 23.6 |
| | en | 1638.2 | ENGLISH: 100.0 | 100.0 |
| | es | 1966.1 | ENGLISH: 94.4, SPANISH: 5.6 | 5.6 |
| | fr | 2062.6 | ENGLISH: 95.2, FRENCH: 4.4, CHINESE: 0.4 | 4.4 |
| | ja | 2162.1 | ENGLISH: 86.8, JAPANESE: 13.2 | 13.2 |
| | ru | 1937.9 | ENGLISH: 98.8, RUSSIAN: 0.8, CHINESE: 0.4 | 0.8 |
| | sw | 3044.8 | ENGLISH: 99.6, SWAHILI: 0.4 | 0.4 |
| | te | 3852.4 | ENGLISH: 98.8, CHINESE: 0.4, TELUGU: 0.8 | 0.8 |
| | th | 1767.3 | ENGLISH: 100.0 | 0.0 |
| | zh | 1438.9 | ENGLISH: 62.0, CHINESE: 38.0 | 38.0 |
| prefix | bn | 4570.1 | BENGALI: 92.8, ENGLISH: 7.2 | 92.8 |
| | de | 1621.8 | GERMAN: 100.0 | 100.0 |
| | en | 1572.2 | ENGLISH: 100.0 | 100.0 |
| | es | 2343.2 | SPANISH: 99.6, ENGLISH: 0.4 | 99.6 |
| | fr | 1354.3 | FRENCH: 99.6, ENGLISH: 0.4 | 99.6 |
| | ja | 2194.3 | JAPANESE: 99.2, ENGLISH: 0.8 | 99.2 |
| | ru | 1470.3 | RUSSIAN: 98.8, CHINESE: 0.8, ENGLISH: 0.4 | 98.8 |
| | sw | 4442.6 | SWAHILI: 85.2, ENGLISH: 14.4, TAGALOG: 0.4 | 85.2 |
| | te | 6041.8 | TELUGU: 90.4, ENGLISH: 9.2, CHINESE: 0.4 | 90.4 |
| | th | 2448.6 | THAI: 88.4, CHINESE: 10.8, ENGLISH: 0.8 | 88.4 |
| | zh | 1149.3 | CHINESE: 100.0 | 100.0 |
| system | bn | 2196.2 | ENGLISH: 99.2, BENGALI: 0.8 | 0.8 |
| | de | 1643.4 | GERMAN: 92.8, ENGLISH: 7.2 | 92.8 |
| | en | 1664.3 | ENGLISH: 100.0 | 100.0 |
| | es | 1766.7 | ENGLISH: 84.4, SPANISH: 15.6 | 15.6 |
| | fr | 1841.7 | ENGLISH: 91.6, FRENCH: 8.4 | 8.4 |
| | ja | 2036.4 | JAPANESE: 17.2, ENGLISH: 82.8 | 17.2 |
| | ru | 1631.3 | RUSSIAN: 49.2, ENGLISH: 50.8 | 49.2 |
| | sw | 2984.6 | ENGLISH: 98.8, SWAHILI: 1.2 | 1.2 |
| | te | 3942.8 | ENGLISH: 91.6, TELUGU: 8.4 | 8.4 |
| | th | 1775.4 | ENGLISH: 91.6, THAI: 8.0, CHINESE: 0.4 | 8.0 |
| | zh | 1295.8 | ENGLISH: 12.0, CHINESE: 88.0 | 88.0 |
| combined | bn | 4823.5 | BENGALI: 98.0, ENGLISH: 2.0 | 98.0 |
| | de | 1479.8 | GERMAN: 100.0 | 100.0 |
| | en | 1553.2 | ENGLISH: 100.0 | 100.0 |
| | es | 2503.3 | SPANISH: 100.0 | 100.0 |
| | fr | 1269.8 | FRENCH: 100.0 | 100.0 |
| | ja | 1941.4 | JAPANESE: 100.0 | 100.0 |
| | ru | 1577.6 | RUSSIAN: 99.6, CHINESE: 0.4 | 99.6 |
| | sw | 4525.5 | SWAHILI: 88.4, ENGLISH: 11.6 | 88.4 |
| | te | 6046.9 | TELUGU: 93.6, ENGLISH: 6.4 | 93.6 |
| | th | 2244.8 | THAI: 92.4, CHINESE: 7.2, ENGLISH: 0.4 | 92.4 |
| | zh | 1118.8 | CHINESE: 100.0 | 100.0 |

as *overthinking* (Liu et al., 2024) where reasoning models expend excessive compute in their long CoTs and lead to worse performance (Cuadron et al., 2025; Chen et al., 2024; Sui et al., 2025).

Table 11: Results for **s1-7B**.

| Strategy | Reasoning Language | avg len | Language Distribution | Language Compliance |
|---|---|---|---|---|
| translated_wait | bn | 3355.8 | ENGLISH: 100.0 | 0.0 |
| | de | 2490.5 | ENGLISH: 68.0, GERMAN: 32.0 | 32.0 |
| | en | 2050.4 | ENGLISH: 100.0 | 100.0 |
| | es | 3156.8 | ENGLISH: 83.6, SPANISH: 16.4 | 16.4 |
| | fr | 2544.1 | ENGLISH: 90.4, FRENCH: 9.2, CHINESE: 0.4 | 9.2 |
| | ja | 3381.9 | ENGLISH: 72.0, JAPANESE: 28.0 | 28.0 |
| | ru | 2742.3 | ENGLISH: 96.0, RUSSIAN: 2.8, CHINESE: 1.2 | 2.8 |
| | sw | 5381.5 | ENGLISH: 100.0 | 0.0 |
| | te | 5232.8 | ENGLISH: 97.6, TELUGU: 2.4 | 2.4 |
| | th | 2654.8 | ENGLISH: 89.6, THAI: 8.4, CHINESE: 2.0 | 8.4 |
| | zh | 1471.1 | CHINESE: 99.2, ENGLISH: 0.8 | 99.2 |
| prefix | bn | 3814.5 | BENGALI: 91.6, ENGLISH: 7.2, CHINESE: 1.2 | 91.6 |
| | de | 2183.4 | GERMAN: 98.0, CHINESE: 1.6, ENGLISH: 0.4 | 98.0 |
| | en | 2405.3 | ENGLISH: 100.0 | 100.0 |
| | es | 3468.3 | SPANISH: 99.2, CHINESE: 0.4, ENGLISH: 0.4 | 99.2 |
| | fr | 1712.5 | FRENCH: 99.2, CHINESE: 0.8 | 99.2 |
| | ja | 4976.8 | JAPANESE: 99.6, ENGLISH: 0.4 | 99.6 |
| | ru | 2242.7 | RUSSIAN: 82.4, CHINESE: 13.6, ENGLISH: 4.0 | 82.4 |
| | sw | 7653.5 | SWAHILI: 55.2, ENGLISH: 43.6, TAGALOG: 1.2 | 55.2 |
| | te | 6649.8 | TELUGU: 82.4, ENGLISH: 17.2, CHINESE: 0.4 | 82.4 |
| | th | 3239.8 | THAI: 80.0, CHINESE: 17.6, ENGLISH: 2.4 | 80.0 |
| | zh | 1675.4 | CHINESE: 100.0 | 100.0 |
| system | bn | 3508.5 | ENGLISH: 96.4, BENGALI: 3.2, CHINESE: 0.4 | 3.2 |
| | de | 2093.8 | GERMAN: 97.2, CHINESE: 2.8 | 97.2 |
| | en | 2721.8 | ENGLISH: 100.0 | 100.0 |
| | es | 4099.5 | SPANISH: 100.0 | 100.0 |
| | fr | 1872.1 | FRENCH: 99.6, CHINESE: 0.4 | 99.6 |
| | ja | 2513.9 | JAPANESE: 100.0 | 100.0 |
| | ru | 2166.5 | RUSSIAN: 79.2, CHINESE: 9.2, ENGLISH: 11.6 | 79.2 |
| | sw | 5310.8 | ENGLISH: 100.0 | 0.0 |
| | te | 5250.4 | ENGLISH: 96.8, TELUGU: 3.2 | 3.2 |
| | th | 3438.9 | THAI: 93.2, CHINESE: 6.8 | 93.2 |
| | zh | 1528.5 | CHINESE: 100.0 | 100.0 |
| combined | bn | 4094.8 | BENGALI: 97.6, CHINESE: 0.8, ENGLISH: 1.6 | 97.6 |
| | de | 2046.7 | GERMAN: 100.0 | 100.0 |
| | en | 2857.6 | ENGLISH: 100.0 | 100.0 |
| | es | 3907.6 | SPANISH: 100.0 | 100.0 |
| | fr | 1657.1 | FRENCH: 99.6, CHINESE: 0.4 | 99.6 |
| | ja | 4804.9 | JAPANESE: 100.0 | 100.0 |
| | ru | 2450.9 | RUSSIAN: 86.4, CHINESE: 13.2, ENGLISH: 0.4 | 86.4 |
| | sw | 7393.2 | SWAHILI: 81.6, ENGLISH: 18.4 | 81.6 |
| | te | 6403.0 | TELUGU: 98.4, ENGLISH: 1.6 | 98.4 |
| | th | 3609.9 | THAI: 92.8, CHINESE: 7.2 | 92.8 |
| | zh | 1734.0 | CHINESE: 100.0 | 100.0 |

Table 12: Performance comparison of different language forcing strategies across multiple model sizes and languages on fixed 8k thinking tokens. Languages are categorized into high-resource (HRL: `de`, `en`, `es`, `fr`, `ru`, `ja`, `zh`) and low-resource (LRL: `bn`, `sw`, `te`, `th`) groups.

| Model | Method | bn | de | en | es | fr | ja | ru | sw | te | th | zh | ALL | HRL | LRL |
|---|---|---|---|---|---|---|---|---|---|---|---|---|---|---|---|
| s1.1-32B | Baseline | 90.8 | **90.8** | 96.0 | **93.2** | 89.6 | 87.6 | 93.2 | 72.4 | 68.4 | **91.6** | 88.0 | 87.4 | 91.2 | 80.8 |
| | translated_wait | **91.2** | 90.4 | 94.8 | **93.2** | 89.2 | **89.2** | 92.0 | **73.2** | **70.8** | **91.6** | 90.0 | **87.8** | 91.3 | **81.7** |
| | prefix | 85.2 | 90.4 | 95.6 | 92.8 | **90.4** | 84.8 | **94.0** | 65.2 | 63.6 | 88.8 | **90.8** | 85.6 | **91.3** | 75.7 |
| | system | 87.6 | 90.0 | **96.4** | 91.2 | 86.8 | 85.2 | 92.8 | 71.2 | 67.2 | 90.8 | 89.6 | 86.0 | 90.3 | 79.3 |
| | combined | 82.8 | 89.2 | 95.2 | 91.6 | 88.8 | 85.2 | 92.8 | 58.4 | 63.2 | 89.2 | 90.4 | 84.3 | 90.5 | 73.4 |
| s1.1-14B | Baseline | **86.8** | 90.4 | 94.4 | **93.6** | 88.4 | 83.6 | 92.4 | 59.6 | 60.0 | 89.2 | 89.6 | 84.4 | 90.3 | 73.9 |
| | translated_wait | 85.6 | 90.0 | **96.8** | **93.6** | 86.8 | **85.2** | 92.4 | **63.2** | 61.2 | 90.8 | 89.2 | **85.0** | 90.6 | **75.2** |
| | prefix | 81.2 | **90.4** | 95.2 | 92.0 | **90.4** | 82.8 | 92.8 | 44.4 | 55.2 | 86.8 | **91.2** | 82.0 | **90.7** | 66.9 |
| | system | 84.0 | 88.8 | 95.2 | 91.2 | 87.2 | 82.8 | 91.2 | 58.8 | **62.0** | 90.4 | 90.4 | 83.8 | 89.5 | 73.8 |
| | combined | 81.2 | 90.0 | 93.2 | 92.0 | 86.4 | 82.4 | 90.4 | 36.8 | 54.4 | 89.2 | 88.4 | 80.4 | 89.0 | 65.4 |
| s1.1-7B | Baseline | **72.0** | **87.6** | 92.4 | 88.8 | 83.2 | **82.4** | 88.0 | 24.0 | 36.8 | 81.6 | **86.0** | **74.8** | **86.9** | 53.6 |
| | translated_wait | 69.2 | 84.0 | 93.2 | 89.2 | **87.2** | 76.4 | 87.2 | 24.0 | **37.2** | 84.0 | 83.2 | 74.1 | 85.8 | **53.6** |
| | prefix | 64.0 | 82.8 | **93.6** | 86.8 | **87.2** | 68.0 | 85.6 | 14.4 | 24.0 | 74.4 | 84.0 | 69.5 | 84.0 | 44.2 |
| | system | 71.6 | 84.0 | 90.8 | **92.0** | 82.8 | 74.4 | **87.6** | **25.6** | 36.8 | 76.8 | 82.0 | 73.1 | 84.8 | 52.7 |
| | combined | 60.8 | 84.0 | 92.8 | 88.0 | 83.6 | 72.8 | 86.4 | 14.8 | 27.6 | 74.4 | 83.6 | 69.9 | 84.5 | 44.4 |

Table 13: Number of average thinking tokens for each reasoning langauge when the MGSM task questions are asked in a particular query language.

| Query Language | Reasoning Language | | | | | | | | | | | AVG |
|---|---|---|---|---|---|---|---|---|---|---|---|---|
| | bn | de | en | es | fr | ja | ru | sw | te | th | zh | |
| bn | 4,559 | 1,974 | 2,190 | 3,473 | 1,640 | 2,276 | 1,977 | 4,900 | 5,461 | 2,852 | 1,710 | 3,001 |
| de | 3,838 | 1,642 | 1,910 | 2,838 | 1,159 | 1,794 | 1,617 | 3,510 | 5,073 | 2,025 | 1,327 | 2,430 |
| en | 3,429 | 1,239 | 1,467 | 2,253 | 1,075 | 1,625 | 1,341 | 3,431 | 5,154 | 1,703 | 1,075 | 2,162 |
| es | 3,736 | 1,388 | 1,779 | 2,512 | 1,225 | 1,665 | 1,467 | 3,600 | 5,323 | 1,875 | 1,133 | 2,336 |
| fr | 3,868 | 1,562 | 1,886 | 2,886 | 1,218 | 2,103 | 1,577 | 4,077 | 5,033 | 2,218 | 1,263 | 2,517 |
| ja | 3,994 | 1,623 | 2,028 | 2,627 | 1,366 | 2,033 | 1,616 | 3,957 | 5,334 | 2,285 | 1,333 | 2,563 |
| ru | 4,105 | 1,392 | 1,713 | 2,695 | 1,170 | 2,007 | 1,469 | 4,055 | 4,818 | 2,132 | 1,098 | 2,423 |
| sw | 5,958 | 2,945 | 2,955 | 3,801 | 2,144 | 4,114 | 3,108 | 3,867 | 6,836 | 3,976 | 2,705 | 3,855 |
| te | 5,761 | 3,341 | 3,528 | 4,160 | 2,473 | 3,574 | 3,337 | 5,934 | 6,277 | 3,879 | 2,557 | 4,074 |
| th | 4,166 | 1,687 | 2,000 | 2,801 | 1,300 | 1,993 | 1,756 | 4,107 | 5,247 | 2,279 | 1,241 | 2,597 |
| zh | 4,017 | 1,650 | 1,737 | 2,521 | 1,365 | 1,822 | 1,649 | 4,233 | 5,344 | 2,009 | 1,117 | 2,496 |
| AVG | 4,311 | 1,858 | 2,108 | 2,960 | 1,466 | 2,273 | 1,901 | 4,151 | 5,445 | 2,475 | 1,505 | ↘2,585 |

Table 14: Performance scores across different reasoning languages given query language on MT-AIME2024 dataset (subset of languages covering high-resource and low-resource languages).

| Query Language | Reasoning Language | | | | | | Range (max - min) |
|---|---|---|---|---|---|---|---|
| | cy | de | en | fr | ml | kn | |
| cy | 10.8 | 39.7 | 41.6 | **42.8** | 21.8 | 8.9 | 33.9 |
| de | 17.7 | **46.1** | 44.5 | 43.4 | 28.6 | 21.7 | 28.4 |
| en | 20.8 | 47.8 | **51.2** | 48.8 | 31.5 | 25.5 | 30.4 |
| fr | 9.8 | **50.9** | 50.3 | 47.3 | 26.9 | 17.7 | 41.1 |
| ml | 8.5 | 42.0 | **45.7** | 38.9 | 18.4 | 10.6 | 37.2 |
| kn | 4.5 | 36.6 | **40.8** | 35.4 | 12.5 | 7.8 | 36.3 |
| AVG | 12.0 | 43.9 | **45.7** | 42.8 | 23.3 | 15.4 | ↘30.3 |

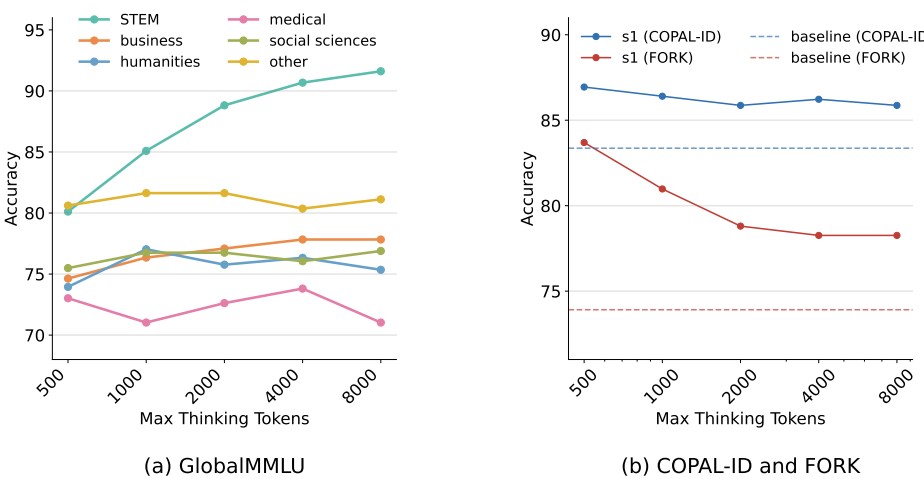

(a) GlobalMMLU

(b) COPAL-ID and FORK

Figure 11: Effects of thinking time for s1 models on different domains of multilingual Global-MMLU benchmark (subfigure (a)) and cultural commonsense knowledge (FORK) and reasoning (COPAL-ID) benchmarks (subfigure (b)). Dashed lines indicates zero-shot prompting of Qwen-32B baseline in (b).

