# OpenReview forum: "Crosslingual Reasoning through Test-Time Scaling"
_ICLR.cc/2026/Conference — Submitted to ICLR 2026_

### Official Review · Reviewer_YbNK · 2025-10-21

**Soundness:** 2
**Presentation:** 3
**Contribution:** 2
**Rating:** 4
**Confidence:** 4

**Summary:**

This paper investigates the extent to which fine-tuning of English reasoning can be generalised across languages. Experimenting with s1 models, the authors find that sequential test-time scaling for English-centric reasoning models through longer CoTs improves multilingual math reasoning.

**Strengths:**

- The paper is clearly written, well-structured, and easy to follow.
- The authors perform experiments on s1 models with different sizes, which show that larger models benefit from crosslingual test-time scaling, which contrasts with contemporary work that draws negative conclusions based on 1.5B models.
- The authors report a language-mixing pattern in the models that quotes non-English phrases related to the question prompts during the thinking process; The authors also introduce several strategies to control the reasoning language of models.

**Weaknesses:**

- The main idea (analyzing test-time scaling across multiple languages) is incremental since it simply extends prior scaling analyses to a multilingual scenario. It would be interesting to see a new analytical framework, or scaling metric; otherwise the contribution appears limited.
- The paper primarily presents empirical scaling trends but does not attempt to uncover or explain the underlying causes behind these patterns. For instance, analysing how factors such as the multilingual capability of the base model (Qwen), language family, and data distribution influence scaling behaviour could provide insight into the mechanisms behind the observed trends. Without such interpretation, the findings tend to be descriptive rather than explanatory, highlighting expected improvements with larger models but offering limited insight into the causes of these trends.
- The study relies solely on the s1 model family for all experiments, which limits the generalizability of the findings and conclusions. In addition, while the paper attributes the performance gain to cross-lingual generalisation, the observed behaviour is that the model generates in English language at least 92.5% of the time; this suggests a strong monolingual bias rather than cross-lingual reasoning. This likely reflects the English-centric nature of the s1 fine-tuning data, which limits the model’s ability to reason and generate coherently across languages, as evidenced in Table 2. The observation in Sec. 6.2 is interesting, where the authors show some unexpected performance comparison of reasoning language; however, since the analysis is based on a single model (and possibly a single model size), it remains unclear whether this finding generalises beyond the specific setup used in the paper.

Others:
- Due to the small scale of AIME 2024 (30 samples), the 5 runs in this paper may not be enough, and the commonly used setting is generally 16 or even larger.
- Line 84: long chain-of-thoughts (long CoTs). The abbreviation has been defined earlier.
- Line 93: lengths g before. Typo?

**Questions:**

See above Weaknesses.

---

> ### Author Response · Authors · 2025-11-25
>
> We thank the reviewer for the positive feedback on our work, and here are our responses to the raised weaknesses:
>
> > **W1: The main idea (analyzing test-time scaling across multiple languages) is incremental since it simply extends prior scaling analyses to a multilingual scenario. It would be interesting to see a new analytical framework, or scaling metric; otherwise the contribution appears limited.**
>
> While our experimental setup is straightforward, we respectfully disagree that addressing misconceptions in the field, as evidenced by [1]'s recognition as an oral paper at ACL 2025, represents merely incremental work. [1] concluded that "test-time scaling may not generalize as effectively to multilingual tasks" and suggested that 70B may be the threshold when transfer happened; and our work shows that the crosslingual transfer via test-time scaling can happen at a smaller scale.
>
> Furthermore, our quote-and-think analysis work contrasts the conclusion from [2] and shows that English-trained reasoning models can mix languages during generation. Lastly, our language forcing study shows a simpler mitigation strategy than [3] that requires training to ensure high language compliance.
>
> Collectively, these findings on English-trained reasoning models challenge established conclusions and provide actionable insights for the community; therefore, we believe our contributions extend well beyond incremental work suggested by the reviewer.
>
> > **W2: The paper primarily presents empirical scaling trends but does not attempt to uncover or explain the underlying causes behind these patterns. For instance, analysing how factors such as the multilingual capability of the base model (Qwen), language family, and data distribution influence scaling behaviour could provide insight into the mechanisms behind the observed trends. Without such interpretation, the findings tend to be descriptive rather than explanatory, highlighting expected improvements with larger models but offering limited insight into the causes of these trends.**
>
> We added the section Appendix B.3 to show the effects of both language family and pretraining data distribution on crosslingual reasoning transfer. Here, we approximate the language representation within Qwen’s pretraining data with that of mC4 pretraining data, since Qwen team did not release any information about the multilingual distribution of their data.
>
> Based on our analysis, we found that:
> - crosslingual transfer from English does not depend on the language family of the target language.
> - base model’s language representation in pretraining is the dominant factor determining crosslingual reasoning transfer performance.
>
>
> > **W3: The study relies solely on the s1 model family for all experiments, which limits the generalizability of the findings and conclusions.**
>
> We experimented on Llama3-8B model and we observe similar crosslingual reasoning transfer as we scale up the number of thinking tokens. We have added the additional result in Figure 4.
>
> We hope the reviewer finds our clarifications address the concerns, and we would be grateful if the reviewer could reconsider their evaluation upward in light of these responses.
>
> —
>
> [1] Son et al. “Linguistic Generalizability of Test-Time Scaling in Mathematical Reasoning”. ACL 2025 (Oral).
>
> [2] Zhang et al. “Multilingual Large Language Models Are Not (Yet) Code-Switchers”. EMNLP 2023.
>
> [3] Marchisio et al. “Understanding and Mitigating Language Confusion in LLMs”. EMNLP 2024.

---

> > ### Comment · Reviewer_YbNK · 2025-11-26
> > **Official Comment by Reviewer YbNK**
> >
> > Thank you for your rebuttal. I decided to retain my score.

---

### Official Review · Reviewer_jitX · 2025-10-24

**Soundness:** 3
**Presentation:** 3
**Contribution:** 3
**Rating:** 6
**Confidence:** 4

**Summary:**

This paper studies whether English-centric reasoning finetuning (long chain-of-thought training in English) can generalize across languages when scaling test-time thinking tokens. Using the s1 family, the authors discuss three research questions: (RQ1) Does sequential test-time scaling help on multilingual math? (RQ2) How do models mix languages while reasoning? (RQ3) Can we force the reasoning language, and does it matter?

The results and analysis show that for models with more than 3B parameters, increasing thinking tokens substantially boosts accuracy on MGSM, MT-AIME2024, and PolyMath. A 14B s1 reaches ~81% average on MGSM (with 8k thinking tokens) and can even beat some models twice its size. The paper also uncovers a dominant "quote-and-think" phenomenon --- the model is likely to reason in English but quotes key non-English phrases from the prompt and interprets them. Finally, the paper discusses language forcing, finding that models reason better and more efficiently in high-resource languages; while reasoning in low-resource languages both underperforms and consumes more tokens, with a strong negative correlation between token count and accuracy.

**Strengths:**

The paper is written clearly, with tightly scoped research questions. It shows that scaling test-time "thinking" tokens reliably boosts accuracy for models with more than 3B parameters. The tested model size is large enough, like the s1-14B model with 8k thinking tokens. The analysis uncovers a mechanistic "quote-and-think" phenomenon in which the model mainly reasons in English but quotes non-English fragments from the prompt. The paper also provides actionable guidance on language control via a simple forcing recipe, demonstrating that high-resource languages yield better accuracy with fewer tokens.

**Weaknesses:**

- (i) The findings are primarily on s1 (basis: Qwen). It would be better to verify if identical trends hold for other multilingual bases (e.g., Llama, DeepSeek-Distilled-R1) under the same setup.
- (ii) Section 6 on "Language Forcing" is also an important section. However, the analysis is only conducted on one dataset, that is, MGSM, limiting the generality of the findings. It is expected to see a similar study on other multilingual reasoning datasets.

**Questions:**

As mentioned above, my main concern about this paper is the robustness of the experiments, including the diversity of tested models and evaluated datasets. Two extra experiments will benefit significantly --- one with Deepseek-R1-Distill, and one with another multilingual reasoning dataset when analyzing "Language Forcing".

---

> ### Author Response · Authors · 2025-11-25
>
> We thank the reviewer for the positive feedback on our work, and here are our responses to the raised concerns:
>
> > **W1: The findings are primarily on s1 (basis: Qwen). It would be better to verify if identical trends hold for other multilingual bases (e.g., Llama, DeepSeek-Distilled-R1) under the same setup.**
>
> We experimented on Llama3-8B model and we observe similar crosslingual reasoning transfer as we scale up the number of thinking tokens. We have added the additional result in Figure 4.
>
> > **W2: Section 6 on "Language Forcing" is also an important section. However, the analysis is only conducted on one dataset, that is, MGSM, limiting the generality of the findings. It is expected to see a similar study on other multilingual reasoning datasets.**
>
> We have added Table 14 as the language forcing results on MT-AIME2024 dataset. We see that similar core findings were observed. For instance, language forcing to generate reasoning in lower-resource languages such as  Kannada (kn) and Welsh (cy) result in substantially poorer reasoning performance.
>
> | Query Language | cy | de | en | fr | ml | kn |
> |----------------|-------|-------|-------|-------|-------|-------|
> | cy | 10.8 | 39.7 | 41.6 | 42.8 | 21.8 | 8.9 |
> | de | 17.7 | 46.1 | 44.5 | 43.4 | 28.6 | 21.7 |
> | en | 20.8 | 47.8 | 51.2 | 48.8 | 31.5 | 25.5 |
> | fr | 9.8 | 50.9 | 50.3 | 47.3 | 26.9 | 17.7 |
> | ml | 8.5 | 42.0 | 45.7 | 38.9 | 18.4 | 10.6 |
> | kn | 4.5 | 36.6 | 40.8 | 35.4 | 12.5 | 7.8 |
> | AVG | 12.0 | 43.9 | 45.7 | 42.8 | 23.3 | 15.4 |
>
> We hope the reviewer finds our clarifications address the concerns, and we would be grateful if the reviewer could reconsider their evaluation upward in light of these responses

---

### Official Review · Reviewer_kB3a · 2025-10-25

**Soundness:** 3
**Presentation:** 3
**Contribution:** 3
**Rating:** 6
**Confidence:** 3

**Summary:**

This paper investigates cross-lingual generalization of English-centric reasoning models. The authors show that longer chain-of-thoughts improve multilingual reasoning, even in low-resource languages, and reveal a “quote-and-think” pattern where models reason in English about non-English inputs. They also find that models perform better when reasoning in high-resource languages. The study highlights both the potential and limitations of English-based reasoning transfer.

**Strengths:**

1. Extensive experiments have been conducted to investigate the extent to which English reasoning finetuning can generalize across languages.
2 .The 'quote-and-think' approach is particularly insightful. It demonstrates that the model does not merely translate non-English input into English before reasoning, but actively parses and reasons over the original linguistic structure.

**Weaknesses:**

I want to know whether the improvement is not only from English to low-resource languages, but whether using any high-resource language similarly improves reasoning in relatively low-resource languages.

**Questions:**

See the weaknesses.

---

> ### Author Response · Authors · 2025-11-25
>
> We appreciate the reviewer’s positive feedback on our work. To address the weakness raised,
>
> > **I want to know whether the improvement is not only from English to low-resource languages, but whether using any high-resource language similarly improves reasoning in relatively low-resource languages.**
>
> We trained our base model Qwen2.5-14B-Instruct on the s1 dataset translated from English into Mandarin Chinese and French, and here’s our average results on 8 low-resource languages in MT-AIME:
>
> | Model | 500 | 1000 | 2000 | 4000 | 8000 |
> |-------|-----|------|------|------|------|
> | Trained in English | 4.1±2.1 | 8.2±3.2 | 14.1±4.5 | 25.9±4.8 | 30.8±5.0 |
> | Trained in Mandarin Chinese | 4.5±2.3 | 7.1±2.8 | 13.6±3.9 | 22.5±3.5 | 31.6±4.7 |
> | Trained in French | 3.4±2.0 | 6.8±2.9 | 14.4±4.2 | 20.4±4.3 | 28.7±4.9 |
>
> We refer the reviewer to Figure 5 for the results with standard deviation plotted.
>
> We hope the reviewer finds our clarifications address the concerns, and we would be grateful if the reviewer could reconsider their evaluation upward in light of these responses.

---

> > ### Comment · Reviewer_kB3a · 2025-11-26
> >
> > Thank you for your rebuttal. I appreciate the clarifications provided and will maintain my score.

---

### Official Review · Reviewer_6ww1 · 2025-11-01

**Soundness:** 3
**Presentation:** 3
**Contribution:** 2
**Rating:** 4
**Confidence:** 3

**Summary:**

This paper studies whether sequential test-time scaling (longer CoTs) learned from English-only reasoning fine-tuning transfers to non-English inputs. Using the s1 family, they find that increasing the thinking token budget improves multilingual math performance. They analyze language mixing, suggesting that models mostly reason in English but follow a “quote-and-think” pattern, performing foreign-language quotation given the non-English inputs. They introduce language forcing, and claim that it reliably switches the language for reasoning, observing that reasoning in high-resource languages is more accurate and more token-efficient than using low-resource languages.

**Strengths:**

1. The research questions are clearly stated and focused, with the experiments neatly mapped / aligned to answering them, and the paper is well-written.
2. It is useful to tie the core behavior to how the model was trained, specifically in the s1 training data, as per appendix C.4 (this should be brought to the main text, in my view).
3. An complete evaluation of the query language and reasoning language in Table 2 is illustrative of the strong performing high-resource languages and the low-resource languages at the bottom in performance.

**Weaknesses:**

1. While it is evident (and plausible) that the “quote-and-think” behavior is present in generations, it is not necessarily causal, as is implied. Is there any evidence to suggest that removing or masking the non-English spans degrades accuracy?
2. Excluding Latin-script languages to avoid misclassification limits generality, so it’s a bit unclear how robust the “quote-and-think” behavior is across the full set of languages.
3. All experiments use s1 / the Qwen family of models, it would be valuable to demonstrate this effect with other families to show a broader trend.

**Questions:**

See weaknesses for some concerns and suggestions.

1. Can you run the MT-AIME experiments multi-seed and report standard deviation? Since it’s a small number of samples, it is important to ensure robustness of the results.
2. Since the forcing prompts might be semantically uneven across languages, have you considered a human / expert study of how natural they are and ablating various translations would be valuable?

---

> ### Author Response · Authors · 2025-11-25
>
> We thank the reviewer for the positive feedback on our experimental results, and here are our responses to the raised concerns:
>
> > **W1: While it is evident (and plausible) that the “quote-and-think” behavior is present in generations, it is not necessarily causal, as is implied. Is there any evidence to suggest that removing or masking the non-English spans degrades accuracy?**
>
> We followed the reviewer’s suggestion and ran the removal experiment where we replaced the phrases within the quotation mark with an empty string and forced s1 to regenerate CoT from the end of the quotation mark. We confirm that removing the non-English spans on those instances which exhibit quote-and-think behaviors leads to a decrease in MGSM accuracy, especially for  lower-resource languages such as Swahili (sw) and Telugu (te).
>
> | | bn | de | en | es | fr | ja | ru | sw | te | th | zh |
> |----------------|--------------|--------------|--------------|--------------|--------------|--------------|--------------|--------------|--------------|--------------|--------------|
> | Before removal | 80.1 ± 1.2 | 84.2 ± 0.9 | 90.2 ± 0.7 | 81.7 ± 1.1 | 79.9 ± 1.3 | 85.4 ± 0.8 | 83.7 ± 1.0 | 53.8 ± 2.8 | 59.1 ± 2.4 | 80.9 ± 1.1 | 85.4 ± 0.9 |
> | After removal | 76.3 ± 1.3 | 83.4 ± 1.1 | 88.4 ± 0.8 | 80.8 ± 1.2 | 78.4 ± 1.4 | 82.3 ± 1.0 | 80.5 ± 1.2 | 47.7 ± 3.1 | 53.6 ± 2.5 | 78.1 ± 1.3 | 82.3 ± 1.1 |
>
> We have added Appendix C.5 for such causal effect analysis.
>
> > **W2: Excluding Latin-script languages to avoid misclassification limits generality, so it’s a bit unclear how robust the “quote-and-think” behavior is across the full set of languages.**
>
> We expanded our quote-and-think analysis to cover all the languages in MSGM and we observed the same behavior. We have replaced the Figure 6 with the new results.
>
> > **W3: All experiments use s1 / the Qwen family of models, it would be valuable to demonstrate this effect with other families to show a broader trend.**
>
> We experimented on Llama3-8B model and we observe similar crosslingual reasoning transfer as we scale up the number of thinking tokens. We have added the additional result in Figure 4.
>
> > **Q1: Can you run the MT-AIME experiments multi-seed and report standard deviation? Since it’s a small number of samples, it is important to ensure robustness of the results.**
>
> We have updated Figure 2 with standard deviation, where we ran with 16 different seeds. We originally omitted the standard deviation (ran with 5 seeds) for clarity of the figure.
>
> > **Q2: Since the forcing prompts might be semantically uneven across languages, have you considered a human / expert study of how natural they are and ablating various translations would be valuable?**
>
> We agreed that an expert study on naturalness would be extremely valuable. We had only done preliminary analysis of the reasoning outputs where we looked at a very small subset of the outputs. We found that CoTs in higher-resource languages like Mandarin Chinese are more fluent, whereas CoTs in lower-resource languages experience different degrees of degeneration (e.g., repeated tokens). A systematic evaluation was beyond our available resources for the study, and we have added this as the future work of our study.
>
> We hope the reviewer finds our clarifications address the concerns, and we would be grateful if the reviewer could reconsider their evaluation upward in light of these responses.

---

### Author Response · Authors · 2025-11-25

### General Response

We sincerely appreciate the reviewers' thoughtful feedback on our work, particularly the recognition of:
1. **Novel insights on crosslingual reasoning transfer**: Reviewers acknowledged our finding that test-time scaling enables effective crosslingual transfer in reasoning models, which challenges prior work suggesting limited multilingual generalization. Several reviewers highlighted the value of our "quote-and-think" analysis, which reveals that models actively parse and reason over the original linguistic structure rather than simply translating to English.
2. **Comprehensive experimental evaluation**: Reviewers commended our systematic evaluation across multiple model sizes, languages, and benchmarks, which demonstrates the robustness and generality of our findings.
3. **Practical actionable insights**: Reviewers appreciated our language forcing strategies and their implications for controlling reasoning behavior in multilingual settings.

Reviewers raised important questions about causality, generalizability across model families and datasets, and the need for additional analysis on factors influencing crosslingual transfer. In response, we have made the following main revisions:
- **Appendix C.5**: We added causal analysis demonstrating that removing non-English spans from quote-and-think instances degrades reasoning accuracy, particularly for lower-resource languages like Swahili and Telugu.
- **Figure 2**: We updated MT-AIME experiments with 16-seed evaluations and standard deviations to ensure robustness.
- **Section 4.3**: We expanded our experiments to the Llama3-8B model family, confirming that similar crosslingual reasoning transfer occurs across different model architectures. We also added experiments showing that training in other high-resource languages (Mandarin Chinese, French) also enables crosslingual transfer.
- **Figure 6 and related analysis**: We extended our quote-and-think analysis to cover all languages in MGSM, not just non-Latin-script languages, demonstrating the generality of this phenomenon.
- **Table 14**: We extended our language forcing analysis to the MT-AIME2024 dataset.
- **Appendix B.3**: We included analysis of how language family and pretraining data distribution affect crosslingual reasoning transfer, revealing that base model language representation is the dominant factor.
We believe these additions comprehensively address the reviewers' concerns while strengthening the empirical foundation and generalizability of our findings.

---

### Meta-Review · Area_Chair_BZeJ · 2025-12-11

**Summary:**

This paper explores the crosslingual generalization of English-centric reasoning models through test-time scaling. Key contributions include demonstrating that longer Chain-of-Thought (CoT) improves multilingual performance and identifying a "quote-and-think" mechanism. Reviewers generally praised the extensive experiments and the novelty of the mechanistic analysis. However, concerns were initially raised regarding the reliance on a single model family (s1/Qwen), the robustness of results, and whether the findings were merely descriptive rather than explanatory.

**Reviewer Concerns:**

The authors provided a comprehensive rebuttal, though the reception varied due to the inherent complexity of the research topic:

- Addressed: The authors made significant efforts to expand the experimental scope by including Llama3-8B, increasing random seeds for robustness, and performing ablation studies to prove the causality of the "quote-and-think" mechanism. However, establishing the definitive underlying mechanisms of crosslingual transfer seems to be an inherently difficult task. Consequently, while the authors have broadened their experimental evidence, this expansion could be insufficient to fully persuade some reviewers (notably YbNK) regarding the depth of the explanatory power and the "why" behind the observed transfer.

- Outstanding: Reviewer YbNK maintained that the work remains incremental and descriptive. Despite the authors adding Appendix B.3 to analyze language families and pretraining data distributions, this reviewer remained unconvinced that the paper sufficiently explains the root causes of the phenomenon.

**Reviewer Scores:**

No reviewer actually raised their score.

- Reviewers kB3a and YbNK explicitly stated they would maintain/retain their original scores (6 and 4, respectively) after reviewing the author's response.
- Regarding the reviewers who did not respond (jitX, 6ww1), considering the conservative reactions of the active reviewers and the challenging nature of the research topic requiring extensive experimental validation, it is highly likely that they would not have raised their scores even if they had fully participated in the discussion.

---

### Decision · Program_Chairs · 2026-01-26

Reject